



# EstSoil-EH v1.0: An eco-hydrological modelling parameters dataset derived from the Soil Map of Estonia

Alexander Kmoch[1], Arno Kanal[1,†], Alar Astover[2], Ain Kull[1], Holger Virro[1], Aveliina Helm[3], Meelis Pärtel[3], Ivika Ostonen[1] and Evelyn Uuemaa[1]

[1]Department of Geography, Institute of Ecology and Earth Sciences, University of Tartu, Vanemuise 46, Tartu, 51003, Estonia
[2]Chair of Soil Science, Institute of Agricultural and Environmental Sciences, Estonian University of Life Sciences, Fr.R. Kreutzwaldi 5, Tartu, 51014, Estonia
[3]Department of Botany, Institute of Ecology and Earth Sciences, University of Tartu, Lai 40, Tartu, 51005, Estonia
† deceased, 07th of May, 2019

Correspondence to: Alexander Kmoch (alexander.kmoch@ut.ee)

**Abstract.** The Soil Map of Estonia is a vector dataset that maps more than 750 000 soil units throughout Estonia at a scale of 1:10 000. It is the most detailed and information-rich dataset for soils in Estonia, a Baltic country with an area of approximately 45 000 km². For each soil unit, it describes the soil type, quality, texture, and layer information with a series of complex text codes. However, to use it as an input for numerical modelling using process-based physical models, these text codes must be

translated into numbers. Various generalisations and aggregations for agricultural soils for less-detailed versions of the map have been made at a scale of 1:100 000 and 1:200 000.

In this study, we create an extended eco-hydrological dataset for Estonia, the EstSoil-EH v1.0 (Kmoch et al., 2019a; doi:10.5281/ZENODO.3473290), containing derived numerical values for the following data in all of the mapped soil units in the 1:10 000 soil map: soil profiles (e.g., layers, depths), texture (clay, silt, sand components), coarse fragments and rock

content, and physical variables related to water and carbon (bulk density, hydraulic conductivity, organic carbon content). Ultimately, our objective was to develop a reproducible method for deriving numerical values to support modelling and prediction of eco-hydrological processes in Estonia using the popular Soil and Water Assessment Tool.

The developed methodology and dataset will be an important resource for the Baltic region. Countries like Lithuania and Latvia have similar historical soil records from the Soviet era that could be turned into value-added datasets such as the one

we developed for Estonia.

## 1 Introduction

The Soil and Water Assessment Tool (SWAT; https://swat.tamu.edu/) has been developed and applied during the past 30 years to evaluate the effects of alternative management decisions on water resources and non-point-source pollution in river basins through the simulation of physical processes (Arnold et al., 1998; Douglas-Mankin et al., 2010). SWAT is widely used

internationally and is increasingly applied in Northern European and Baltic watersheds to better assess the hydrological state of the environment based on modelling of the most relevant physical processes (Piniewski et al., 2018; Tamm et al., 2016,



2018). However, a main input factor for SWAT is detailed soil data, which does not exist for many countries on national scale or which exists with insufficiently fine spatial resolution. In addition, it is complicated to derive the values of the model parameters.

At the global level, two main soil databases are available. The first was made available by the United Nations Food
and Agricultural Organisation (FAO) through its Soils Portal: the Harmonized World Soil Database (HWSD) v1.2 (Fischer et al., 2008; http://www.fao.org/soils-portal/soil-survey/soil-maps-and-databases/harmonized-world-soil-database-v12/en/). The dataset resulted from a collaboration between FAO and Austria's International Institute for Applied Systems Analysis (http://www.iiasa.ac.at/), ISRIC–World Soil Information (https://www.isric.org/), the Institute of Soil Science of the Chinese Academy of Sciences (http://english.issas.cas.cn/), and the Joint Research Centre of the European Commission
(https://ec.europa.eu/info/departments/joint-research-centre_en). HWSD is a 30-arc-second raster database with more than 15000 different soil mapping units. It combines existing regional and national updates of soil information from around the world, including SOTER (https://www.isric.org/explore/soter), ESD (https://esis.sc.egov.usda.gov/Welcome/pgReportLocation.aspx?type=ESD), the Soil Map of China (https://esdac.jrc.ec.europa.eu/content/soil-map-china), and WISE (https://www.isric.org/explore/wise-databases). It also
contains information from the 1:5 000 000-scale FAO-UNESCO Soil Map of the World (FAO, 1990; http://www.fao.org/soils-portal/soil-survey/soil-maps-and-databases/faounesco-soil-map-of-the-world/en/).

The other global-level soil dataset is SoilGrids250m, which provides global gridded soil information based on machine learning (Hengl et al., 2017) and is made accessible via an interactive Web interface (https://soilgrids.org/) with sophisticated standards-based data access via the OGC Web Coverage Services
(https://www.opengeospatial.org/standards/wcs). SoilGrids250m provides values for sand, silt, clay, and rock fractions, and organic carbon and carbon stocks at several depths, which can be used as inputs for SWAT. At a regional level, the European Soil Database v2.0 (Panagos et al., 2012; https://esdac.jrc.ec.europa.eu/content/european-soil-database-v20-vector-and-attribute-data) is the only harmonized soil database for Europe, and also covers Estonia. It contains the soil geographical database SGDBE (vector data), which includes a number of essential soil attributes, and an associated database (PTRDB), with
attribute values that have been derived through pedotransfer rules. The European database also includes the Soil Profile Analytical Database, which contains measured and predicted soil profiles for Europe as well as soil organic carbon (SOC) projections for Europe that include 26 European countries at a resolution of 1 km. Wösten et al. (1999) developed a database of HYdraulic PRoperties of European Soils (HYPRES).

Existing national-scale soil datasets that have been developed to be used by SWAT currently only exist for the United
States. Cordeiro et al. (2018) developed an official soil dataset for SWAT for Canada. Apart from these efforts, no consistent methodology has been used to develop soil datasets at national, continental, or global scales so that the data is optimised for use in SWAT (Batjes, 1997; Dobos et al., 2005).

In Estonia, systematic large-scale soil mapping was launched in 1949, with agronomy students assisting (Estonian Landboard, 2017; "mullakaardi_seletuskiri.pdf"). Starting in 1954, a special survey was carried out under the supervision of



the Ministry of Agriculture. Aerial photographs were used as the basis for this activity. By 1992, Estonia's soil cover had been mapped by the Soil Survey Department of the former Institute of Estonian Agroprojects at a scale of 1:10 000. In addition to inspecting arable land, forests, and other land types between 1989 and 1991, the remaining former Soviet military areas were also mapped. Between 1997 and 2001, the soil map was digitized and attribute data was inserted into the database, resulting

in the official National Soil Map of Estonia as a vector dataset that mapped 750 000 soil units at a scale of 1:10 000 (Estonian Landboard, 2017; https://geoportaal.maaamet.ee/est/Andmed-ja-kaardid/Mullastiku-kaart-p33.html). It is the most detailed and information-rich dataset for soils in Estonia and, to the best of our knowledge, is also the most detailed national-scale digital soil database in the world. For each soil unit, it describes the soil type, quality, texture, and layer information using a series of complex text codes. However, to use it as an input for process-based models such as SWAT, these text codes must

be translated into numbers. Processing a soil dataset into a format readable by SWAT is a time-consuming process because not all data required by SWAT are readily available (Bossa et al., 2012; Rahman et al., 2012). Various datasets have been created that generalise values for agricultural soils in Estonia to produce less detailed versions at scales of 1:100 000 and 1:200 000 (Kõlli et al., 2009; Tamm et al., 2018). However, no large-scale high-resolution soil database is currently compatible with SWAT.

The objective of the present study was to develop a reproducible method for deriving numerical values as inputs for modelling and for predicting hydrological processes with SWAT in Estonia. By developing this method, we aimed to develop a fully numeric soil database for Estonia. In this study, we derive numerical values for the key characteristics for the whole Soil Map of Estonia at a 1:10 000 scale for soil profiles (e.g., layers, depths), textures (clay, silt, and sand contents), coarse fragment and rock content, and physical variables related to water and carbon (bulk density, hydraulic conductivity, SOC) for

all of the mapped soil units. The format is also suitable for use with the ArcSWAT graphical user interface (https://swat.tamu.edu/software/arcswat/) for the SWAT model. We conclude with a discussion of the short-comings and uncertainties of the developed dataset.

Prévost (2004) described predictions of soil properties from the SOC content, and found that SOC was closely related to soil bulk density (BD) and porosity. Suuster et al. (2011) emphasized the importance of BD as an indicator of soil quality,

site productivity, and soil compaction and proposed a PTF for the organic horizon in arable soils.

Van Looy et al. (2017) reviewed existing PTFs and documented the new generation of PTFs that have been developed by different disciplines of Earth system science. They emphasized that PTF development must go hand in hand with suitable extrapolation and upscaling techniques to ensure that the PTFs correctly represent the spatial heterogeneity of the soils. Abdelbaki (2018) evaluated the predictive accuracy of 48 published PTFs for predicting BD using State Soil Geographic

(STATSGO) and Soil Survey Geographic (SSURGO) soil databases from the United States. They also proposed and validated a new PTF for predicting BD using SOC inputs.

However, reliable estimates for SOC have been difficult to obtain due to a lack of global data on the SOC content of each soil type (Eswaran et al., 1993). Very few SOC datasets are available for countries or regions. For example, the Northern Circumpolar Soil Carbon Database (Tarnocai et al., 2009; https://bolin.su.se/data/ncscd/) was developed to describe the SOC





pools in soils of the northern circumpolar permafrost region. SOC stocks were also predicted under future climate and land cover change scenarios using a geostatistical model for predicting current and future SOC in Europe (Yigini and Panagos, 2016). Kõlli et al. (2009) published estimates of the SOC stocks for forests, arable lands, and grasslands and for all of Estonia. They constrained their finding by noting that their estimates were calculated based on the mean SOC stock for each soil type

and the corresponding area in which the soil type was distributed. Putku (2016) used the large-scale Soil Map of Estonia at the polygon level for SOC stock modelling for mineral soils in arable land of Tartu county. SOC and soil-hydraulic predictions for Estonia need to consider that Estonia is located relatively far north and hosts large areas of peatlands.

In this study, we derived numerical values for the following data in all of the mapped soil units in the 1:10 000 soil map: soil profiles (e.g., layers, depths), texture (clay, silt, sand components, and coarse fragments), rock content, and physical

variables related to water and carbon (organic carbon content, bulk density, hydraulic conductivity, available water capacity and erodibility factor). We present the development of a reproducible method for deriving numerical values from a Soil Map of Estonia to support modelling and prediction of eco-hydrological processes with the popular Soil and Water Assessment Tool and we create an extended ready-to-use dataset containing the additional parameters.

## 2 Materials and Methods

### 2.1 Pre-processing and screening of the initial soil database

The Soil Map of Estonia is a vector layer for geographical information system software that can be downloaded from the Republic of Estonia Land Board Web site (https://www.maaamet.ee/en) in several formats under a permissive open data license. A copy with the original shapefile dataset, the related required documentation and checksums has been archived for reference (Estonian Landboard, 2017; https://datadoi.ut.ee/handle/33/103). The soil map contains the following attribute fields:

-   Soil type: a designation of the soil name

-   Texture: texture classes defined by fine and coarse fragments, and to which depth the same texture is observed (layer)

-   Potential fertility: estimated, prospective fertility that can be achieved after land improvement, this includes the potential fertility for wetlands or peatlands if they were drained

-   Organic horizon thickness

-   Rockiness: the type: i.e. material (e.g. limestone or sandstone rock), shape (e.g. flat, round, blocky) and % of the soil volume occupied by rocks (stones with a diameter $\geq$ 20 cm, in $m^3$/ha, in the upper 30 cm of the soil, expressed with a degree of hardness)

-   Overall soil type group: categories numbered from 0 to 22

Most attributes are encoded as "string" values, which include both letters and numbers. Exceptions are fertility and the

generalised soil type group, which are stored as integer values. The important fields soil type and texture, are not stored as standardised class values, but are instead a coded description based on abbreviations that are then combined with numbers for example depths and indictors for level of erosion, and are grouped together for different depths within the same attribute field.



These description-based attribute values make it difficult to derive the foundational numerical values for sand, silt, clay and coarse fragments from the codes and to make them more consistent and usable in calculations and statistical analyses. In addition, our data screening revealed that the attribute values sometimes contradict the official legend for the Soil Map of Estonia. For example, the soil type reference sheet provided with the soil map lists ca. 120 different soil types in Estonia (Estonian Landboard, 2017; "muldade_tabel.pdf") and the soil legend document describes 9 main texture classes and 12 soil skeleton types, i.e., coarse fragments and rock morphology (Estonian Landboard, 2017; "mullalegend.pdf"). However, the database's attribute table contains 7067 unique variations for soil types, which resulted from the use of many specific local derivatives and transcription errors. Similarly, the texture column actually contains 87240 unique values instead of 9, 21 (9+12) or 108 (9x12). Considering the possible permutations of these soil types and textures, it would be prohibitively difficult to develop any kind of reasonable standardisation for the soil parameters before cleaning and unifying the dataset. Therefore, we performed extensive database standardisation on the original Soil Map of Estonia as the working basis and derive all further variables based from the standardised dataset sequentially. Figure 1 illustrates the four major working packages to derived the desired eco-hydrological parameters. The subsequent sections are structured accordingly.

## 2.2 Analysis of soil type and texture codes and extraction of basic physical and textural values

### 2.2.1 Standardising soil types

We used the main soil types from the soil type reference sheet that accompanies the Soil Map of Estonia to standardise the soil type fields in the spatial dataset and added several widely-used soil types that were not in the original reference list. We developed a short algorithm in Python to find the best match from the soil type reference list (Kmoch et al., 2019b; "01_soilmap_soiltypes_textures_layers.ipynb"). The algorithm progressively shortens the name from the right and compares the results with the "Soil type" field in the database. If no match is found, it then tries to find the name in a list of known exceptions. We created a lookup table (Kmoch et al., 2019a; "soil_types_error_rules_lookup.xlsx") that captures more than 300 entries that provide a soil type substitute code from the extended 135 soil types from the soil reference list to these not automatically identifiable soil types. The soil types and the Estonian soil names were then related to the FAO World Reference Base (WRB) soil codes (FAO, 2015) after the data have been corrected and standardised for each map unit in the extended soil dataset. An exemplary except is demonstrated in Table 1. The finalised table of the standardised 135 main soil types is provided as supplemental spreadsheet (Kmoch et al., 2019a; "soil_types_legend.csv").

### 2.2.2 Pre-processing texture codes

The Soil Map of Estonia's "texture" field encodes the texture and general soil layer structure for each mapped soil unit in a structured, rule-based format (based on old Soviet-era paper maps). To derive meaningful numerical values for texture and other soil variables from the soil map, we developed a computer program that encodes these rules into a computer-readable grammar. In addition, we provided a lookup table for wrongly encoded texture codes and historical data-entry errors. The





program provides a complete internal data structure that represents the analysed grammatical representation, which can be evaluated and used to generate numerical values for a variety of variables.

The main implementation of this program is based on the Python library "Arpeggio" (Dejanović et al., 2016; https://pypi.org/project/Arpeggio/), which is a recursive-descent parser based on parsing expression grammars (also known as

the Packrat parser; http://bford.info/packrat/). This let us express rules and symbols (i.e., the grammar) in such a way that our software could parse arbitrary text and find the various definitions of the texture in the same way as the rules are described in the map legend handout.

Listing 1 provides an example of a parsing grammar. At the start of the program, the basic elements are defined, starting with the 9 main fine-textured soil types: "*plsl, pl, tsl, tls, dk, sl, ls, s, l*". The parser honours the order of their definition.

Without these ordered rules, the parser will never find the more complex expression "*plsl*", because it would stop as soon as it encounters the "*pl*" part of the name. We also defined the skeleton types, i.e. coarse fragments, and peat land soils.

The function "*def fine_textured(): return Optional(kPlus), fine-textured_list, Optional(amplifiers), Optional(depth_range)*" demonstrates the flexibility of how a parsing expression grammar parser can be configured. The function can find even optional (0 or 1) elements such as prefixes or suffixes within an arbitrary text stream.

Subsequently, several separators and special indicators must be defined that can precede or be appended in combinations to the abovementioned soil type elements. These were often formatted as subscripts or using special characters. This proved to be a major source of data-entry errors, encoding mistakes, and ambiguities, which had to be handled via additional error-checking code, e.g., lookup tables which are provided in the supplemental materials (Kmoch et al., 2019b; "soil_lib/LoimisLookups.py").

The mapped soil units also encode variations in the soil profile within a given soil unit. Thus, we must differentiate between a vertical separator for the observed soil layers, and a horizontal separator. However, we only considered the vertical component (soil horizons). In addition, these discrete vertical layers are only based the description in the original texture code. To capture and fully evaluate the possible texture codes, it was necessary to capture the meaning of any additional (rare) horizontal separators.

Because there are various data-entry errors and other ambiguities in the actual codes in the soil map dataset, we manually analysed all codes that could not be successfully evaluated by the grammar. Manual inspection was particularly required for codes that did not conform to the general rules described in the original soil legend handout. A full list of non-logical expressions, data-entry errors, and other grammar expressions that could not be easily or usefully standardised is provided as a supplemental Excel spreadsheet (Kmoch et al., 2019a; "texture_error_lookup.xlsx").

The parser for the defined grammar builds a data structure that can be evaluated for physical numerical parameters such as layers, depth, and the sand, silt, clay, and rock contents. This data structure is a Python dictionary object, i.e. a lookup table with nested key-value pairs that hold the parameters and the found values. In the example in Listing 2, it becomes apparent that there is a "/" vertical layer separator (at the bottom, the "code" parameter shows the original texture code for this soil unit), and that depths and fine fractions are accessible separately from the data structure. If a skeleton fraction were defined in the





texture code, then additional to the fine earth information, an additional "constituent" (the skeleton) would be part of the respective layer (i.e. the "soilparts" object). The complete parser Python module is provided as supplemental material (Kmoch et al., 2019b; "soil_lib/LoimisGrammarV2.py").

### 2.2.3 Deriving depths and layers

5   In Subsection 2.2.2, we processed the textual code descriptions to compile the exact Estonian texture types in a standardised and readily available data structure. A base assumption is that most soils in Estonia were sampled to a depth of 1 m, as this is the case for a default soil profile. If larger or smaller depth information was encoded in the original soil texture code, then this would be used for the overall depth of that soil sample. The corresponding parameter required by SWAT is "NLAYERS", which represents the number of soil layers. For each of the layers, we can also read the analysed depth from the

10 soil surface to the bottom of each layer. The SWAT parameters are named SOL_Z# (layer 1-4). Knowing NLAYERS and depth per layer (SOL_Z#), we can derive the maximum soil depth (SOL_ZMX), which represents the maximum depth of the soil profile (mm). We eliminated additional soil parts from the dataset if their resulting layer thickness would be zero. An additional pragmatic decision was made to exclude cumulative vertical soil parts if their depth could not be reliably inferred. For example, "*sand/loamy sand*" indicates two layers, separated by "/". The base assumption is that the profiles have been

15 sampled to the depth of one metre when no additional depth information is available. Therefore, for the given example, no depth information is available for the second layer ("*loamy sand*"). In these cases, we decide to drop the second layer and assign the full depth of 1m for the first layer "*sand*". Another example is shown in Listing 2, where the first layer depth is indicated as a range of 70-110 cm. In this case to derive a single number, the average will be taken resulting in 90 cm for the first layer. The remaining 10 cm filling up to 1m can be assigned to second layer. The Soil Map of Estonia holds depths in

20 centimetres, whereas SWAT requires depths in millimetres. It was therefore also necessary to convert the depths from cm to mm in this step.

   In addition to the evaluation of layer and depth values we assign the extracted Estonian fine earth type and the related USDA texture class per layer in the variable EST_TXT# (layer 1-4, Estonian texture class) and LX_TYPE# (layer 1-4, USDA texture class). This step was conducted as part of the script in the supplemental materials (Kmoch et al., 2019b;

25 "01_soilmap_soiltypes_textures_layers.ipynb").

### 2.2.4 Deriving sand, silt, and clay fractions and rock content

   The foundational numerical values for fine earth and coarse fragments fractions of soil are now derived from the extracted processed and translated texture classes. The USDA soil taxonomy and World Reference Base soil classification systems use 12 textural classes, which are defined based on the sand, silt, and clay fractions (Ditzler et al., 2017). However,

30 the USDA system defines fine particles as having a diameter $\leq 2$ mm, whereas the Soviet-era maps use a diameter of $\leq 1$ mm. The Soviet soil classification also mostly ignores the silt fractions, and focuses on the clay fraction ($\varnothing \leq 0.001$ mm).

   SWAT requires data for the numerical texture input parameters for each layer as follows:





- SOL_CLAY# (layer 1-4): clay content (% soil weight)
- SOL_SILT# (layer 1-4): silt content (% soil weight)
- SOL_SAND# (layer 1-4): sand content (% soil weight)
- SOL_ROCK# (layer 1-4): rock fragment content (% total weight)

Based on the available analysis data and its structure, we derived meaningful numerical texture values using a lookup table that represents our best efforts to account for the size difference between the USDA and Soviet systems and lack of silt data in the Soviet system. There are many finely scaled texture classes in the Estonian system. We assigned USDA texture classes and defined the sand, silt and clay fractions. Table 2 shows examples of the rules. For each texture code, the table provides the combination of sand, silt, and clay contents. In additional we introduced two more classes beyond the well-known USDA

textures classes, i.e. "PEAT" and "GRAVELS". The former states that this soil unit is a peatland, where the peat layer thickness is at least 30 cm. For hydrological modelling reasons we decided to still assign sand, silt and clay fractions to these units in order to provide a continuous hydrological soil surface. To soil units with the class "PEAT" a high clay content was assigned in order to represent the low vertical conductivity at the bottom theses peat bogs. However, for applications that critically evaluate clay content for soil units, the additional "PEAT" texture class (in the LX_TYPE1-4 variable) can be used to apply

additional rules to mask these soil units accordingly. The latter class "GRAVELS" is intended to demark soil units or discrete layers therein, where only a coarse fragment type but no fine textures have been coded in the original texture codes. In these cases, depending on the type of the coarse fragment the layer can consist of gravels, large rocks or massive rock.

The rock content parameter in SWAT does not directly match the soil skeleton descriptions in the Soil Map of Estonia. The soil map reference guide describes stone shapes and sizes, and ranges of these values, using a subscript indicator number

that is often (but not always) appended to the skeleton code. We derived an overall percentage volume of rock based on that indicator number. Table 3 shows how we derived the rock content from the coarse fragments indicator that we obtained from the soil map encoding. This step was also conducted as part of the script in the supplemental materials (Kmoch et al., 2019b; "01_soilmap_soiltypes_textures_layers.ipynb"). This first and fundamental step concluded with a set of variables for each mapped soil unit that include now separate standardised Estonian and English/USDA texture classes per soil layer, number

and depths of layers of the mapped soil unit and numerical values for fine earth and coarse fragments fractions per layer. The complete    workflow    is    coded    in    the    supplemental    materials    (Kmoch    et    al.,    2019b; "01_soilmap_soiltypes_textures_layers.ipynb").

### 2.2.5 Evaluation and validation of extracted texture values

For validation, we used a manually "decoded" part of the Estonian Soil Map for Tartu county. Tartu County covers

about 10% of Estonia and offers a representative subset of the data, as it includes many different soil types, peat bogs, forest, and arable land. It contains 83 364 records. Several members of our research group cleaned and standardised the data on soil types, textures, and depth ranges over the course of several months and collated the results in a spreadsheet. We then compared the software's results with the manual classification results. Each soil unit in question was interpreted by at least two experts,



and when their classifications differed, they discussed the difference until they achieved consensus about the correct classification.

We used two other sources of cross-validation to confirm the accuracy of the derived values: SoilGrids250m. First, we loaded and averaged the raster layers from the SoilGrids250m data for the sand, silt, and clay contents for the top layer.
Next, we rasterized the discrete polygons from the Soil Map of Estonia, including the newly generated extended values, for the same attributes provided by SoilGrids250m to the same raster resolution. For each parameter, we subtracted the SoilGrids250m value from our value to assess the magnitude of the deviation. We observed strong similarity in the general patterns. However, the variance ranged from 30 to 50%. One main cause for this high variation is the scale difference between the discrete polygon values in our data and the more continuous distribution of raster values in the SoilGrids250m dataset.

**2.3 Adding topographic variables as predictor variables**

For the subsequent step of SOC prediction via the Random Forest machine-learning model, we calculated mean, median and standard deviation of several topographic and environmental variables as additional predictor variables. Topographic variables slope, Topographic Wetness Index (TWI), Terrain Ruggedness Index (TRI), and LS-factor were all calculated by using SAGA-GIS software based on a digital elevation model (Conrad et al., 2015). The LiDAR-based Digital
Elevation Model with resolution 1 m was obtained from Estonian Land Board.

**2.3.1 Topographic Wetness Index (TWI)**

The TWI is a topo-hydrological factor proposed by Beven and Kirkby (Beven and Kirkby, 1979) and is often used to quantify topographic control on hydrological processes (Michielsen et al., 2016; Uuemaa et al., 2018) which also are relevant in the soil evolution. TWI controls the spatial pattern of saturated areas which directly affect hydrological processes at the
watershed scale. Manual mapping of soil moisture patterns is often labor-intensive, costly, and not feasible at large scales. TWI provides an alternative for understanding the spatial pattern of wetness of the soil (Mokarram et al., 2015). It is a function of both the slope and the upstream contributing area:

$$TWI = \ln \left( a/_{\tan b} \right) \qquad (1)$$

where a is the specific upslope area draining through a certain point per unit contour length (m$^2$ m$^{-1}$), and b is the slope gradient (in degrees).

**2.3.2 Terrain Ruggedness Index (TRI)**

TRI reflects the soil erosion processes and surface storage capacity which again is relevant from soil evolution perspective. The TRI expresses the amount of elevation difference between neighboring cells, where the differences between the focal cell and eight neighboring cells are calculated:



$$TRI = Y[\textstyle\sum\left(x_{ij} - x_{00}\right)^2]^{1/2} \qquad (2)$$

where $x_{ij}$ is the elevation of each neighbor cell to cell (0,0). Flat areas have a value of zero, while mountain areas with steep ridges have positive values.

### 2.3.3 LS-factor

The potential erosion in catchments can be evaluated using LS factor as used by the Universal Soil Loss Equation (USLE). The LS factor is length-slope factor that accounts for the effects of topography on erosion and is based on slope and specific catchment area (as substitute for slope length). In SAGA-GIS the calculation is based on (Moore et al., 1991):

$$LS = (n + 1)\left(A_s/22.13\right)^n\left(\sin\beta/0.0896\right)^m \qquad (3)$$

where n=0.4 and m=1.3.

### 2.3.4 Drainage area per mapped soil unit

In addition, we calculated the area per mapped soil unit in $m^2$ (area_drain) and in percent of area, which is under drainage (drain_pct). The drainage regimen considered both underground tile drainage and ditch based drainage systems. The drainage information used for this was compiled based on the Estonian Topographic Data Set (ETAK) and the official register

of drainage systems (https://portaal.agri.ee/avalik/#/maaparandus/msr/systeemi-otsing) managed by the Agricultural Board of Ministry of Rural Affairs of Estonia. All the variables were calculated using the GIS software packages QGIS and SAGA.

### 2.4 Predicting Soil Organic Carbon (SOC) and Bulk Density (BD)

The main information described in the Soil Map of Estonia is the soil type and the soil texture. However, soil hydraulic properties and SOC data are needed for many different applications in soil hydrology, and especially for SWAT. Pedotransfer

functions (PTFs) have proven to be useful to indirectly estimate these parameters from more easily obtainable soil data (Van Looy et al., 2017). Therefore, several soil parameters like soil organic carbon, bulk density and saturated hydraulic conductivity must be derived via PTFs and other data assimilation methods. To apply PTFs and other data-assimilation methods, third-party datasets can be used as secondary sources. In the previous steps we have prepared a wide set of input variables, including the numerical fractions for the textural properties, standardised classes for soil type and soil textures, and additional topographic

variables, which we can apply as predictor variables to model the value distribution for SOC and BD. The SWAT model defines these two extended soil physical input parameters as follows:

-   SOL_CBN# (layer 1-4): **organic carbon content** in % soil weight
-   SOL_BD# (layer 1-4): **moist bulk density** in Mg/m³ or g/cm³





In order to map the spatial distribution of SOC in Estonia a machine learning model random forest (RF) was used to predict SOC based on parameters derived from the soil map. RF was preferred to more advanced ML algorithms (e.g., neural networks) because it has shown to be relatively resilient towards data noise and not require preliminary hyperparameter tuning (Breiman, 2001; Caruana and Niculescu-Mizil, 2006). In addition, feature importances can be extracted from the model to determine the most influential predictor variables.

For training, we used measurements of soil organic matter (SOM) or soil organic carbon (SOC) from forest areas (samples sizes: n=100), 4 datasets of samples from Estonian open and overgrown alvars and grasslands (n: 94, 137, 146, 69), peatlands (n=175) and from arable soils transects (n=8964) resulting in 3373 distinct point locations (Kriiska et al., 2019; Noreika et al., 2019; Suuster et al., 2011). Where necessary, the SOM values were translated into SOC via: $SOC = SOM / 1.724$. Many samples from peatlands and arable fields were often sampled within the same mapped soil unit. For these soil units (polygons) the respective soil measurement data was averaged and joined to the respective soil units to reduce the bias of the prediction. After joining the sample size reduced to the 397 distinct training samples for machine learning (Figure 2).

This data was then randomly split into training (60%) and test (40%) sets. An RF regression model was fitted to the training set using the *RandomForestRegressor* function from the Scikit-learn Python library. The model was evaluated by predicting SOC based on the predictor variables of the test set. Finally, the model was applied to soil map polygons without available SOC measurements to predict SOC content in Estonian soils.

Subsequently, we calculated soil bulk density based on texture values and predicted soil organic carbon for each layer in each mapped soil unit polygon, with following PTF (Adams, 1973; Kauer et al., 2019):

$$BD = 1 / ( 0.03476 \times SOM + 0.6098 ) \qquad (4)$$

where: $SOM = SOC \times 1.724$

The workflow of SOC and BD calculation is coded in Python and provided in the supplemental materials (Kmoch et al., 2019b; "03_SOC_RF_preps.ipynb").

## 2.5 Assimilation of additional hydrological variables

In order for this dataset to be more useful in eco-hydrological modelling we develop and add two additional hydrological variables. Saturated hydraulic conductivity ($K_{sat}$) relates soil texture to a hydraulic gradient and is quantitative measure of water movement through a saturated soil. In addition to the ability of transmitting water along a hydraulic gradient we also add available water capacity (AWC) as a variable. AWC describes the soil's ability to hold water and quantifies how much of that water is available for plants to grow. For SWAT these two variables are defined as follows:

- SOL_K(layer 1-4): **saturated hydraulic conductivity** (mm/hr)
- SOL_AWC(layer 1-4): **available water capacity** of the soil layer (mm $H_2O$/mm soil)

We calculated $K_{sat}$ using the improved Rosetta3 software, which implements a pedotransfer model with improved estimates of hydraulic parameter distributions (Zhang and Schaap, 2017). It is based on an artificial neural network (ANN) for the estimation of water retention parameters, saturated hydraulic conductivity, and their uncertainties. For each standardised





texture class, we used the numerical fine earth fractions for sand, silt and clay as inputs for the Rostta3 software and calculated $K_{sat}$ for each layer in each mapped soil unit polygon. We provide a copy of the Rosetta program in the supplemental materials (Kmoch et al., 2019b; "Rosetta-3.0").

In order to calculate available water capacity, we summarized the field capacity (FC, at −330 cm matric potential −0.03 MPa) and wilting point (WP, at −15,848 cm matric potential −1.5 MPa) variables of the 7 soil depths of the EU-SoilHydroGrids 250m resolution raster datasets (Tóth et al., 2017) for each mapped soil unit for the provided depths of 0, 5, 15, 30, 60, 100, and 200 cm. The available water capacity is then calculated for each of the 7 depths by a raster calculation: AWC = FC - WP (Dipak and Abhijit, 2005). The resulting 7 AWC raster layers are then averaged into the respective depth ranges for each of the discrete layers of the Estonian mapped soil units. The Python code of the process for the extraction of FC and WP from the EU-SoilHydroGrids is provided with the supplemental materials (Kmoch et al., 2019b; "05_hydrogrids_extents_and_awc_extract.ipynb").

## 2.6 Calculating the USLE K erodibility factor

To account for erosion, we used the universal soil loss equation (USLE) soil erodibility ($K$) factor (USLE_K) based on the exact values for clay, silt, sand, and rock contents, modified by accounting for the mean SOC content. SWAT requires USLE_K for the top layer of the soil. To calculate USLE_K values for Estonia, we used the previously derived texture classes and SOC content. It was also necessary to provide a soil structural class ($s$) related to the texture, which is defined in the European Soil Database as majority of particles of matter that fit through a sieve of the given size:

- $s = 1$ means very fine granular (1 to 2 mm) - G (good)
- $s = 2$ means fine granular (2 to 5 mm) - N (normal)
- $s = 3$ means medium or coarse granular (5 to 10 mm) - P (poor)
- $s = 4$ means blocky, platy or massive (> 10 mm) - H (humic or peaty top soil)

As in previous sections, we used the Estonian texture code to derive the value using a simple decision tree based on the texture types (1 or 2), amount of rocky material (2 to 3), and the presence of peat soils (4).

Next, we defined soil permeability classes ($p$, Table 4) and the related saturated hydraulic conductivity ranges to derive a permeability class for each soil unit, which became an input for the final USLE_K equation. These classes function similarly to the SWAT hydrologic groups and were derived for major soil textural classes, where $p$ ranges from 1 (very rapid) to 6 (very slow).

We defined the $p$ value for the USLE_K using based on the already predicted hydraulic conductivity $K_{sat}$. We then calculated the required term for an intermediate factor ($M$) that accounts for the balance among the silt, sand, and clay contents:

$$M = (m_{silt} + m_{sand}) \times (100 - m_{clay}) \qquad (5)$$

where $m$ represents the % fraction of weight for the silt, sand, and clay contents (respectively). $K$ is then calculated from $M$ using the following equation:

$$K = \{(2.1 \times 10^{-4} \, M^{1.14}) \times (12 - om) + 3.25(s - 2) + 2.5(p - 3)\} / 100 \qquad (6)$$



where *om* represents organic matter, which we fill in from the predicted SOC (SOC×1.724). The joint process of loading calculated $K_{sat}$ values from the Rosetta3 output, averaging and assimilating the AWC values based onn the FC and WP datasets from the EU-SoilHydroGrids and consequently, calculating USLE_K is described and provided in the supplemental materials (Kmoch et al., 2019b; "02_soilmap_update_Ksat_AWC_USLE-K.ipynb").

## 3 Results

In this study, we developed a Python module that is capable of analysing, extracting, and standardising the soil type and soil texture data from the official Soil Map of Estonia into a reusable, reproducible soil dataset that uses World Reference Base and FAO soil classes and texture descriptions. Figure 4 shows a map of the classified topsoil texture classes derived from the original Estonian texture codes. In addition, it shows the peat soils that cover up to 20% of Estonia, and are an important soil type in such northern countries.

To make such soil information usable in an eco-hydrological modelling context, we derived numerical values for each of the soil units. These values include the number of discretized soil layers (NLAYERS) - a maximum of 4 separate vertical distinct soil layers where described in the original texture codes –the depth of each layer (SOL_Z1-4), and the maximum depth of the sampled profile for each mapped soil unit (SOL_ZMX). Based on the layer information and the extract texture classes we could define the percent fractions per volume of sand (SOL_SAND1-4), silt (SOL_SILT1-4), clay (SOL_CLAY1-4), and coarse fragments (SOL_ROCK1-4) per layer. Figure 5 shows the percent fractions for sand, silt, clay and coarse fragments for the top layer.

### 3.1 Validation of soil type and texture classes extraction and standardisation

For the main soil types, we achieved 97.7% agreement between the software's result and the manual classification. The manual verification of the validation revealed several re-labelling issues from the error lookup table. A visual assessment by two soil sciences senior research staff asserted that the level of similarity of the soil types that were selected by the automated process were closely related. However, the mismatches (1943 records, equivalent to 2.3% of the total records) indicated that the soil experts tended to interpret "errors" based on personal knowledge that may not be reproducible in a strictly automated fashion. For example, some landforms (e.g. eroded material filling low slopes or collapsed cliffs) were originally classified as exceptions to the general classification rule based on the local knowledge of the landscape. When standardising these expert interpretations with the same more general soil type, we reduced the number of mismatched soil type identifiers to 0. Furthermore, it should be emphasised that humans tend to make mistakes when performing repetitive procedures. Therefore, we consider the high accuracy (97.7%) to be a very good result.

For the validation of textures, we used several steps. First, given the high agreement between the software-generated codes and the human-generated codes, we accepted the software's texture codes for use in our subsequent evaluations. Next, we compared the extracted main texture for each layer with the manually coded value:



- 77 870 of 83 364 records (93.4%) showed identical parsing of the full texture code
- 71 635 of the records (85.9%) showed identical interpretation of the first layer's texture type (10 312 records were differently coded, and 1417 produced "no value" errors, in which either the source or validation dataset contained no value, preventing a comparison with the other dataset's value)
- 65 000 of the records (78.0%) showed identical interpretation of the second layer's texture (with 2325 differently coded textures, and 16 038 "no value" errors, of which 15 461 occurred in the automated processed new dataset, and only 577 occurred in the validation dataset)
- 82 507 of the records (99.0%) showed identical interpretation of the third layer's texture (with most errors caused by a non-existent third layer, 334 differently coded, and 523 with a "no value" error)

Our manual assessment of the mismatches indicated the same problem that occurred with the soil types. The expert assessments aimed to keep as much information as possible available in their decoded classification, and this did not always agree with the automated processing rules. Furthermore, the complexity of the Estonian texture rules and the reliance on human judgement creates high uncertainty in some cases, even for human interpretation. In addition, to derive the grammar rules, we added a few simplifying elements, such as omitting some rarely used additional information in the soil texture descriptions. For

example, the Estonian rules allow specification of several soil parts, but as a horizontal distribution within the same mapped soil unit rather than as vertical layers. This is understandably complex, making it difficult to classify this variable soil as a single soil unit. Consequently, it is inevitable that some of these descriptions will not agree with the software's classification.

### 3.2 SOC prediction and validation of Random Forest model

We also calculated several extended soil properties, i.e. *SOC* content and *BD*. The RF regression model was implemented with

the RandomForestRegressor function from the Scikit-learn Python library. The model was evaluated by predicting SOC based on the predictor variables of the test set for the 60:40 split. Figure 3 illustrates the cross-validation scatterplots of observed vs. predicted SOC values for the test/validation sample splits. Following characteristics are reported for the chosen RF model:

- coefficient of determination ($R^2$) score: 0.69
- score of the training dataset with out-of-bag estimate (oob score): 0.58
- Pearsons *r correlation coefficient*, training: 0.90, validation: 0.83

RF feature importances, top 6:

- Clay content (SOL_CLAY1): 0.65
- Terrain Roughness Index, standard deviation (tri_stdev): 0.04
- Sand content (SOL_SAND1): 0.03
- LS-factor, median (ls_median): 0.028
- Area under drainage in percent (drain_prct): 0.027
- Coarse fragments rock content (SOL_ROCK1): 0.024





Figure 6 shows the predicted values of SOC and BD for the top layer. On visual inspection the spatial distribution for the SOC content matches comparatively well with known agricultural areas, where low carbon content prevails, as well as with the peat land areas, which have a very high carbon content.

### 3.3 Extended variables results

Based on the variables derived in previous steps, we could also calculate soil hydraulic parameters, such saturated hydraulic conductivity ($K_{sat}$) and available water capacity (AWC). Figure 7 shows the spatial distribution of $K_{sat}$ andAWC for the first layer. The predicted $K_{sat}$ values for peat areas and wetlands also corresponds with ranges reported in the literature for the sand-silt-clay ratios provided (Gafni et al., 2011). The USLE $K$ erodibility factor for each soil unit was also calculated (Figure 8). We compiled all parameters into a dataset that can now be easily used with SWAT or other eco-hydrological and

land-use-change models. As we are not changing the general geometry or underlying spatial data model of the original soil map, all parameters are only added to the existing mapped soil units and thus, all original soil polygons remain discernible. The dataset (Kmoch et al., 2019a; doi:10.5281/ZENODO.3473290) and the software codes (Kmoch et al., 2019b; https://zenodo.org/record/3473210) have been deposited online.

### 4 Discussion and Future Work

The Soil Map of Estonian is a valuable resource for hydrological, ecological and agricultural studies. It is widely used in Estonia. But before our analysis, a large amount of the dataset's information was not readily usable beyond the field or farm-scale because of the need to manually interpret the specialised soil types and the complexity of the rules that describe the texture or other characteristics of the soil units. We used a multi-step approach to derive a generalised and standardised numerical dataset that will have many potential applications for users of Estonian soil information, including support of

economic, agricultural, or environmental management and input for decision-making and to support more reproducible scientific research based on Estonian soil information. Until the present analysis, numerical and process-based eco-hydrological modelling with tools and models like SWAT or the Regional Hydro-Ecologic Simulation System (RHESSys) were greatly hampered by the need to manually derive useful values from the Soil Map of Estonia to support the modelling.

One challenge in terms of validation is that the datasets we used for validation are informed to some extent by previous

reports about Estonian soil characteristics that are not necessarily more accurate than the results of our classification. Although we accounted for this problem by providing additional comparisons, the scale mismatch between continuous raster datasets and polygon-based data inevitably introduced errors and trade-offs into the comparison. One solution to these problems would be to perform supplemental field sampling to ground-truth the source data and confirm the accuracy of our model's classification based on the field data.

A direct interpretation on the derived discrete layer information as soil horizons needs should not be generalized but checked on per case basis. From the point of end-user, the first layer is not a default 30 cm deep top soil layer. All physical,





chemical and hydraulic properties are based on the analysis of the original texture code per mapped soil units and the resulting discrete layers per unit. This is an important usage constraint, for example in sense of biological activity, as 30 cm soil layer is most active, but for each soil unit it needs to be checked which layers extend into which actual depths. Also the *SOC* content and *BD* are not modelled in a vertical continuum but per discrete values per unit and layer. However, fertile soils, like Luvisols

contain a lot of *SOC* also in deeper layers. But such additional expert knowledge is not encoded in original Soilmap of Estonia, nor in the processing algorithms that derived the extended parameters for this newly generated dataset. However, such additional knowledge, as well as more appropriate models for peatland areas, could be included as additional rules in a subsequent improvement of this dataset. The original purpose of this dataset was to derive values for hydrological modelling purposes and at the same time to stay as close to the original data as possible. From that perspective peat soil units are currently

modelled with assumptions to have a similar behaviour to clay hydrologically. Therefore, the spatial distribution of clay percentage in particular, but also the concurrent physical fractions of sand and silt do not make scientific sense for these areas where peat is prevalent. In order to make the dataset as useful as possible and to identify peatland areas, we introduced the additional class "PEAT" into the USDA classification. While sand, silt, clay and rock content are directly derived values from the original texture codes, *SOC* and $K_{sat}$ are modelled via statistical machine-learning algorithms, which include additional

uncertainty. This should be considered when evaluating *BD* and USLE *K*, which are calculated using *SOC* as an input variable.

      The only variable which we did not model based in dependence of already modelled parameters is *AWC*. Here we summarised the EU-SoilHydroGrids 250m (Tóth et al., 2017) raster datasets for *FC* and *WP* as inputs an external data integration. This is not ideal and can be considered a trade-off between introducing too much uncertainty and an external un-related data source.

20       In the future, we foresee step-wise improvement of our software by developing better PTFs to estimate parameters and to better integrate the presence of peat soils and other specific landscapes and environments in Estonia. Furthermore, statistical machine-learning or neural network and deep learning methods could be tested in order to improve soil classifications and express more complex relationships between soil types and textures. Currently, one specificity of the newly created EstSoil-EH dataset is its discrete nature, as we are only adding derived numerical variables to the existing mapped soil units

(polygons). We do not predict a continuous surface in this study, thus, comparisons with continuous surface parameters predicitons such as in SoilGrids (Hengl et al., 2017) or EU-SoilHydroGrids (Tóth et al., 2017), are not directly possible. However, the workflow could potentially be extended also for creating continuous surface. With appropriate modification (e.g., to use the soil characteristic codes more consistently for a different country), our methodology could also be applied in other countries such as Lithuania or Latvia that share similar historical land- and soil surveying practices.

**Code and data availability.**

The described "EstSoil-EH v1.0" dataset including all supplemental tables and figures is deposited on Zenodo, doi:10.5281/ZENODO.3473290 (Kmoch et al., 2019a). Supplemental software and codes that were used, e.g. the texture-code

parsing scripts, the machine learning model and the parameter calculation Jupyter notebooks are maintained on GitHub (https://github.com/LandscapeGeoinformatics/EstSoil-EH_sw_supplement/releases) and were also deposited on Zenodo, doi:10.5281/ZENODO.3473210 (Kmoch et al., 2019b). The original National Soil Map of Estonia (https://geoportaal.maaamet.ee/est/Andmed-ja-kaardid/Mullastiku-kaart-p33.html ) was archived for reference on the
DataCite- and OpenAire-enabled repository of the University of Tartu, DataDOI, doi:10.15155/re-72 (Estonian Landboard, 2017).

**Author contributions.**

A. Kmoch designed the experiments and code. A. Kmoch, E. Uuemaa, A. Astover, A. Kanal validated soil types and texture values. E. Uuemaa completed terrain analysis and statistics. A. Kull, A. Helm, M. Pärtel, I. Ostonen cleaned and organized the
input SOC datasets. H. Virro developed the SOC RF machine-learning code and experiment. A. Kmoch and E. Uuemaa drafted the manuscript and created the figures, and all authors contributed to the paper writing.

**Competing interests.**

The authors declare that they have no conflict of interest.

**Special issue statement.**

This article is part of the special issue "Linking landscape organisation and hydrological functioning: from hypotheses and observations to concepts, models and understanding". It is not associated with a conference.

**Acknowledgements.**

**Financial support.**

This research has been supported by the Marie Skłodowska-Curie Actions individual fellowships under the Horizon 2020 Programme grant agreement number 795625, the Mobilitas Pluss Postdoctoral Researcher Grant numbers MOBJD233 and PRG352 of the Estonian Research Council (ETAG), the European Regional Development Fund (Centre of Excellence EcolChange), and by the Estonian Environmental Investment Centre.



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

**Figure 1: Flowchart for executed processing steps grouped into the four major work packages**



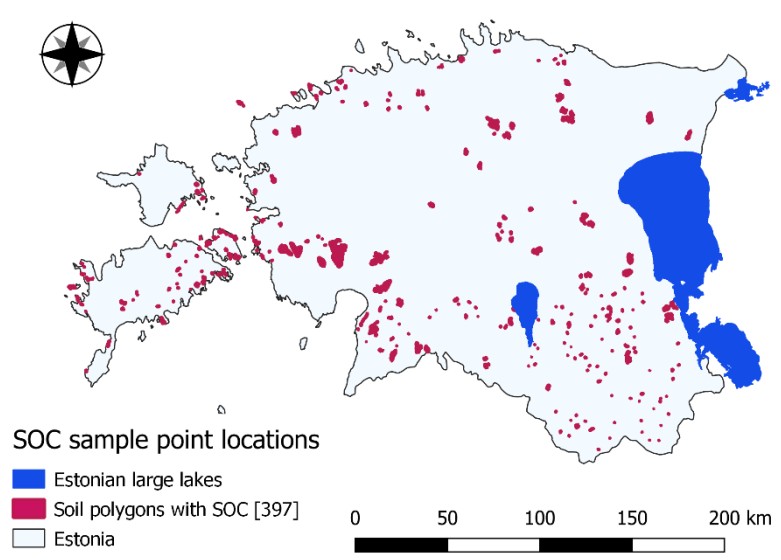

**Figure 2: Distinct soil unit polygons including all sampling locations for the ML training sample.**

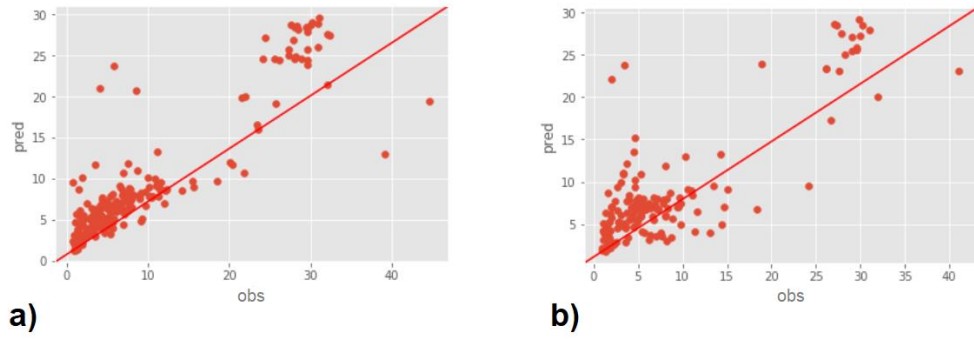

5    **Figure 3: Random Forest model cross-validation scatterplot of observed vs. predicted SOC values for the test/validation sample splits: a) training subsample and b) validation subsample**

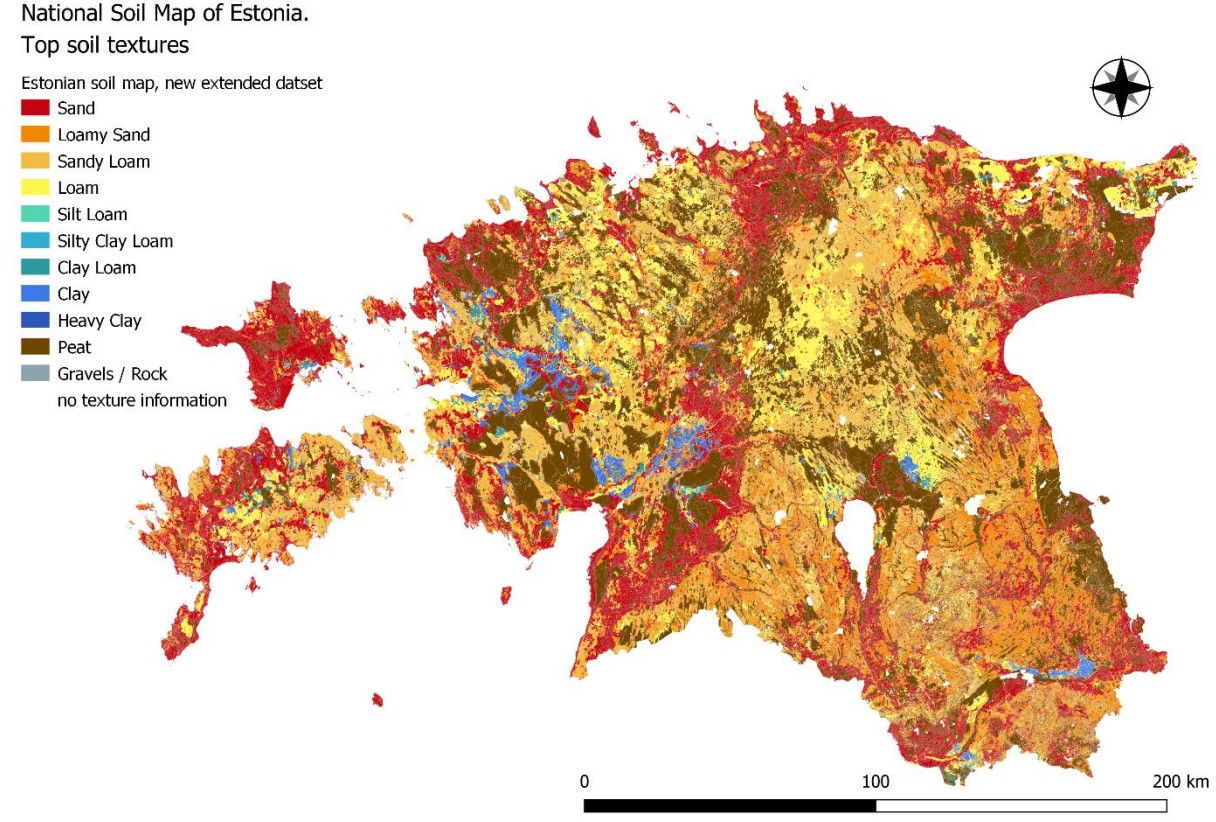

**Figure 4: USDA topsoil textures derived from the original Estonian texture codes by the software developed in the present study, including additional classes "PEAT" and "GRAVELS".**

**Figure 5: Physical soil properties: assigned soil texture fractions of a) sand, b) silt, c) clay, and d) coarse fragments in the first soil layer based on texture classes.**




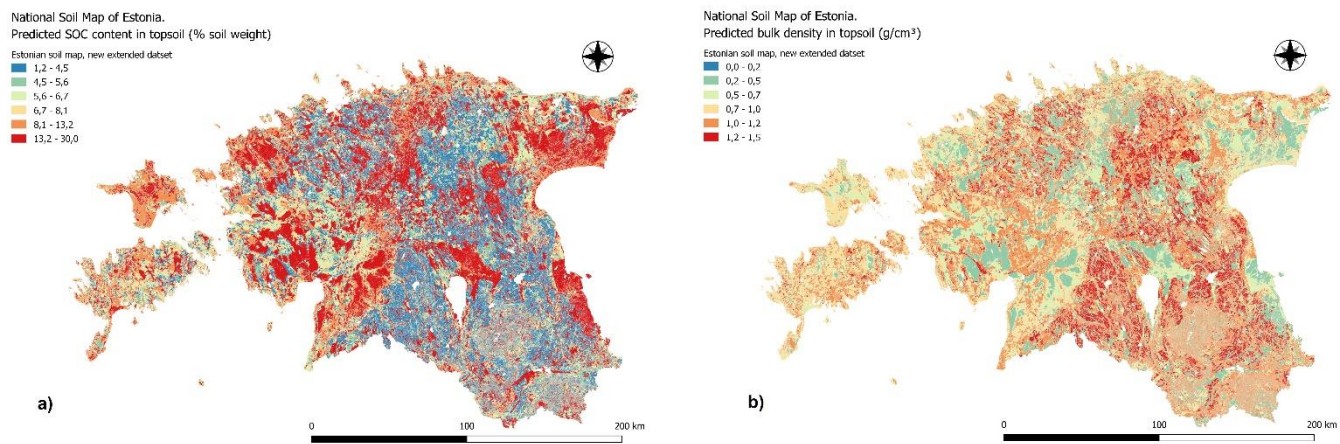

**Figure 6: Extended physical soil parameters: a) predicted soil organic carbon (SOC) and b) bulk density (BD) of the first layer.**

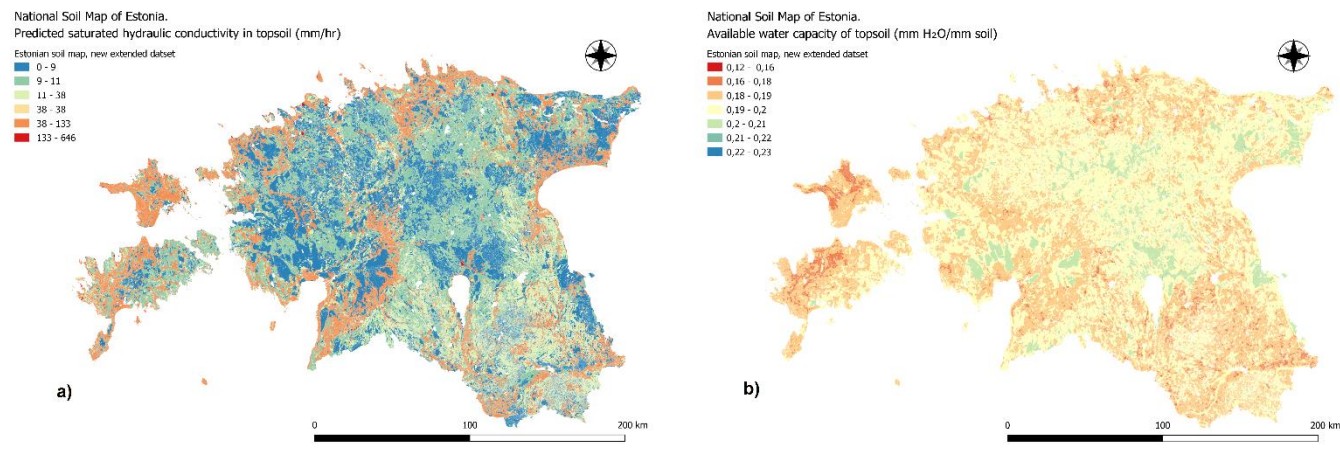

**Figure 7: Soil hydraulic parameters: a) saturated hydraulic conductivity (K_sat) and b) available water capacity (AWC) in the first layer.**

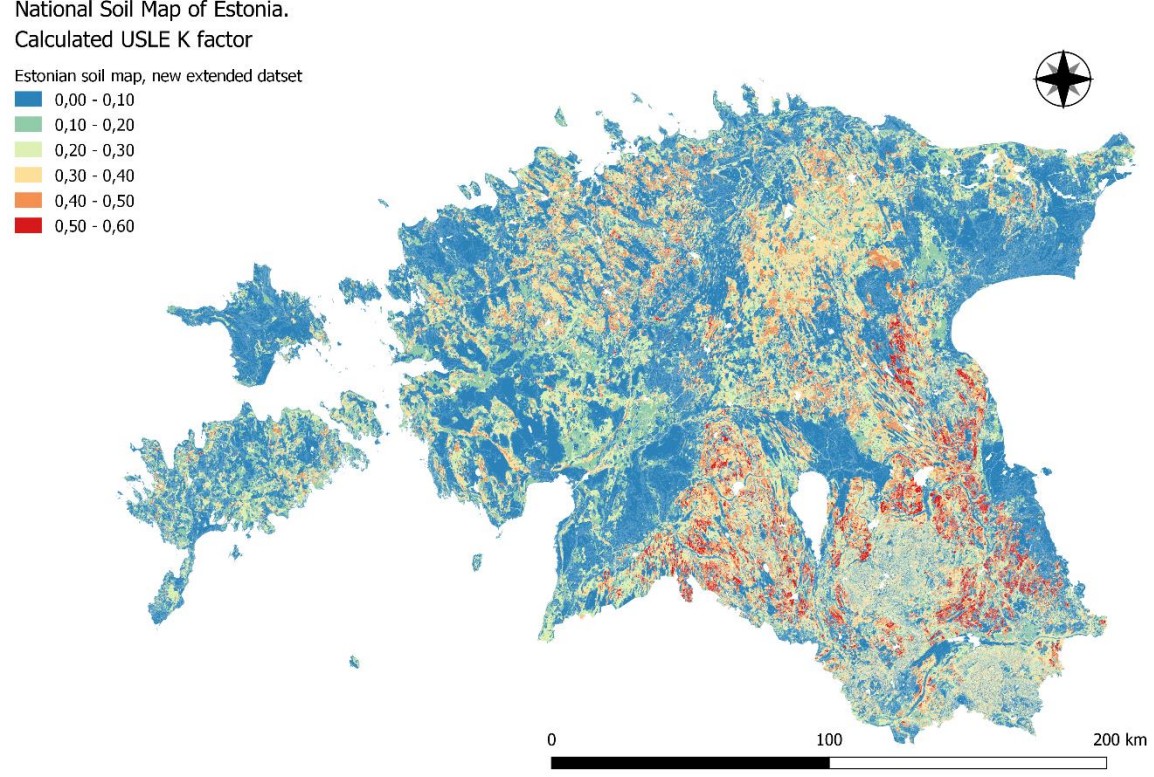

**Figure 8: Calculated USLE *K* erodibility factor based on the numerical parameters derived from the Soil Map of Estonia dataset.**





**Table 1: Examples of the final list of standardised soil types and the added English WRB classes, full list as supplemental spreadsheet ("soil_types_legend.csv")**

| Estonian soil code | Estonian name | Scientific English | WRB_code |
|---|---|---|---|
| Ag | Gleistunud lammimuld | Endogleyic Fluvisols | FL-gln |
| AG | Lammi-gleimuld | Gleyic Fluvisols | FL-gl |
| AG1 | Lammi-turvastunud muld | Histic Fluvisols | FL-hi |
| AM' | Väga õhuke lammi-madalsoomuld | Rheic Sapric Histosols (fluvic) | HS-sa.rh-fv |
| AM" | Õhuke lammi-madalsoomuld | Rheic Sapric Histosols (fluvic) | HS-sa.rh-fv |
| Dg | Gleistunud deluviaalmuld | Endogleyic Umbrisols (deluvic, novic) | UM-gln-del.nv |
| E2I | Keskmiselt erodeeritud kahkjas leetunud ja leetunud muld | Dystric Regosols | RG-dy |
| E2k | Keskmiselt erodeeritud rähkmuld | Epicalcaric Regosols | RG-cap |
| E2o | Keskmiselt erodeeritud leostunud ja leetjas muld | Eutric Brunic Regosols | RG-br.eu |
| E3I | Tugevasti erodeeritud kahkjas leetunu ja leetunud muld | Dystric Regosols | RG-dy |
| E3k | Tugevasti erodeeritud rähkmuld | Epicalcaric Regosols | RG-cap |
| E3o | Tugevasti erodeeritud leostunud ja leetjas muld | Eutric Brunic Regosols | RG-br.eu |

5 **Table 2: Example of the basic rules for deriving numerical values for texture (sand, silt, and clay contents) from the Estonian texture codes and assigned new English and USDA texture classes. These rules were selected by the authors. The full table is provided as a supplemental Excel spreadsheet ("texture_rules_lookup.xlsx")**

| Estonian texture code | Estonian name | English name | USDA texture code | Proportion (%) of total weight | | |
|---|---|---|---|---|---|---|
| | | | | Sand | Silt | Clay |
| l | Liiv | sand | S | 90 | 5 | 5 |
| $l_1$ | sõre liiv | coarse sand | S | 95 | 5 | 0 |
| $l_2$ | sidus liiv | fine sand | S | 90 | 3 | 7 |
| sl | saviliiv | loamy sand | LS | 82 | 9 | 9 |
| $sl_1$ | saviliiv | loamy sand | LS | 82 | 9 | 9 |
| ls | liivsavi | loam | L | 55 | 30 | 15 |
| $ls_1$ | kerge liivsavi | sandy loam | SL | 65 | 20 | 15 |
| $ls_2$ | keskmine liivsavi | loam | L | 55 | 30 | 15 |
| s | Savi | clay | C | 25 | 30 | 45 |



**Table 3: The relationship between the coarse fragments (rock content and shape) indicator from the soil map encoding and the rock content as a % of the total volume. We used the average of each defined range as the singular value required by the SWAT model**

|  | Scale of conversion for rock content | | | | | |
| --- | --- | --- | --- | --- | --- | --- |
| "Skeleton" indicator number | 1 | 2 | 3 | 4 | 5 | 6 |
| Inferred rock content (% of volume) | 6 | 15 | 25 | 40 | 60 | 85 |

**Table 4: Permeability classes used in the USLE_K equation as suggested by the SWAT documentation.**

| Permeability class ($p$) | Texture | Saturated hydraulic conductivity (mm h$^{-1}$) |
| --- | --- | --- |
| 1 (fast and very fast) | Sand | > 61.0 |
| 2 (moderately fast) | Loamy sand, sandy loam | 20.3–61.0 |
| 3 (moderate) | Loam, silty loam | 5.1–20.3 |
| 4 (moderately slow) | Sandy clay loam, clay loam | 2.0–5.1 |
| 5 (slow) | Silty clay loam, sand clay | 1.0–2.0 |
| 6 (very slow) | Silty clay, clay | < 1.0 |



```
# ... at the first the basic elements are defined, starting with the actual soil texture fine-textured soils ("peenes") types
def l(): return 'l'  # liiv, en: sand
def pl(): return 'pl'  # pl - peenliiv, en: fine sand (täiendina peenliivakas)
def plsl(): return 'plsl'  # plsl - peenliivakas saviliiv, en: fine clayey sand
# ... other " finely textured" types
def fine_textured_list(): return [plsl, pl, tsl, tls, dk, sl, ls, s, l]
def fine_textured(): return Optional(kPlus), fine_textured_list, Optional(amplifiers), Optional(depth_range)

# turfs, aka, peat bogs and similar are specially handled types
def t(): return 't'  # t - turvas, en: peat
def th(): return 'th'  # th 15 or th 15-20 humus thickness
def turfs(): return Optional(kPlus), [th, t], Optional(amplifiers), Optional(depth_range)

# soil skeleton ("koores") types, for stone content and shape in the soil part
def kr(): return 'kr'  # kr - kruus, en: gravel
def r(): return 'r'  # r - rähk, en: grit, rubble
# ... and other " skeleton " types
def skeleton_list(): return [kr, kb, pk, ck, lu, v_0, k_0, r_0, r, v, k, p, d]
def skeleton(): return Optional(kPlus), skeleton_list, Optional(amplifiers), Optional(depth_range)

# ... several separators and special indicators can precede or be appended in combinations with the above soil elements
def depth_number(): return RegExMatch(r'\d+')  # at least two digits for depth numbers (well might as well be one)
def depth_range(): return Optional(kPlus), depth_number, ZeroOrMore('-', depth_number)
def vertiSep(): return '/'
def horiSep(): return ';'

# ... and eventually get composed into aggregated lists of components per encoded profile
def constituent(): return ZeroOrMore(skeleton), ZeroOrMore([fine_textured, turfs, Optional(alternateComma)])
def paraSeq(): return OneOrMore(constituent)
def soilParts(): return paraSeq, ZeroOrMore(vertiSep, paraSeq)
def texture_grammar(): return OneOrMore(soilParts, sep=horiSep), EOF
```

**Listing 1: Examples of grammar definitions. The grammar is built upon the defined basic combinations of text and symbols that are found in the database, and comprises the rules for processing those encodings. The full grammar is presented in the supplemental materials** (Kmoch et al., 2019b; "soil_lib/LoimisGrammarV2.py" Python module)

```
{'type': 'loimis',
 'count': 2,
 'soilparts': [{'count': 1,
                'paraseq': [{'count': 1,
                             'constituents': [{'type': 'peenes',
                                               'karbonaat': False,
                                               'code': 'l',
                                               'amp': False,
                                               'depth': {'range': True, 'from': 700.0, 'to': 1100.0}}]}]},
               {'count': 1,
                'paraseq': [{'count': 1,
                             'constituents': [{'type': 'peenes',
                                               'karbonaat': False,
                                               'code': 'ls',
                                               'amp': 2,
                                               'depth': False}]}]}],
 'code': 'l70-110/ls₂'}
```

**Listing 2: The data structure that results from a computed grammatical evaluation. At the bottom, the original texture code for this soil unit is shown.**