# Peer review of "EstSoil-EH: An eco-hydrological modelling parameters dataset derived from the Soil Map of Estonia"

_Earth System Science Data, 2019_

## Referee Comment (RC1) · Anonymous Referee #1 · 26 Nov 2019

The manuscript of Kmoch et al. describes a methodology for deriving high resolution 3D soil property data of Estonia, which is published with the manuscript. The data basis for the methodology is the National Soil Map of Estonia and the soil properties are derived with a special focus on the parameters necessary for running the SWAT model. These parameters include the saturated hydraulic conductivity, field capacity, wilting point and the USLE K erodibility factor. Such large-scale soil data is highly valuable for soil hydrological and water quality modeling on a scale relevant for decision makers (e.g., national scale). Organizing, homogenization and distribution of soil properties on such a large scale is very challenging and I acknowledge the work the authors did here. However, there are some points that prevent the manuscript from being published at

the current state.

General comments

1) Structure: The manuscript is very technical and includes too many details. The main step is the transformation of a text based soil classification (Soil Map of Estonia) to soil texture, which is then (together with SOC, bulk density and topographic information) used for deriving the soil hydraulic properties. All the details (especially the grammar definition parts) makes it difficult to follow these main steps. The explanation of all codes for transforming the letter codes to texture could e.g. be a part of the dataset itself as a documentation.

2) Texture: The step of transforming a text coded soil classification into a numerical texture value is a very crucial point of the methodology. All the focus on the "grammar definition" hides the main step of the transformation, which is done with Table 2. However, it is nowhere cited or mentioned how this table was derived. Is it based on the literature or on own data? This table is the main factor influencing your final results, hence it should be carefully described how you come up with this values. Furthermore, this rises the general point of the missing validation of your final soil texture product (that then influences the hydraulic properties). You only validate your grammar-generated codes and do an "expert check". However, the texture itself is not validated with measured data (as far as I understand). You mention on page 9 line 3-9 that you validated it against SoilGrids250m, but it is important to show this validation. An expert check alone is not enough, since other user of your data cannot assess the uncertainty. You need a reliable texture database for validating your results and hence Table 2 (and Table 3). For your SOC prediction you show such a validation and you correctly mention the relevance of validation of your other data on page 15 line 27-29.

3) Data quality: A data paper should be supportive for the dataset and help the users to evaluate the data and its quality (e.g. uncertainties). This is missing at the moment and instead of focusing on the grammar methodology you should rather present your final

derived hydraulic data with e.g. appropriate diagrams. This includes the uncertainties derived from the texture + SOC classification and also the uncertainty introduced by the pedotransfer functions you used (here ROSETTA). This is also relevant for me as a referee. At the moment for me it is really difficult to evaluate your data in a feasible time. You also mention the problems to derive USDA texture from the old soviet-era based texture system, which ignores the silt fraction and has a different definition for the gravel-sand boundary (page 7 line 29-31). This of course includes a lot of uncertainty, but I understand the benefits of transforming the texture to the often used USDA classification (e.g. usability of pedotransfer functions). I suggest to also include the soviet texture into your data. This can help to evaluate the error introduced by the two different systems and potentially allows to use the data with another "texture transfer function" (different from Table 2).

4) Dataset check: By checking randomly sampled polygons in the final GIS product (.shp) I recognized some problems with the soil layers. E.g. FID 96775 has two layers with SOC and bulk density values are shown in layer 1 and 2. However, texture values are indicated in layer 1 and 3, whereas layer 2 is empty. Similar problem was found in FID 178514 with only one layer but texture values in layer 1 and 2. Please check your data again.

5) SWAT focus: The manuscript focuses too strongly on SWAT. Although the dataset was created for using it with the SWAT model, this is not important in the data paper. Of course you can mention that the presented data is enough for many modeling purposes (e.g. SWAT), but at the moment the focus on SWAT makes the manuscript difficult to understand. E.g. on page 7 line 6-15, just mention that you have defined different layers.

6) Highlight the need for your dataset: You mention similar global or regional datasets (page 2 line 4-28). However you miss to highlight the need of your dataset. What is different from the others or "better" in your dataset? Why it needs a new dataset? For calculating the available water capacity you use the dataset of Tóth et al. (2017) which

is not mentioned in this section. Why is this dataset not usable for parameterizing models in Estonia?

In summary the manuscript should rather focus on the quality of the data than on the methodology of the grammar definition. That does not mean that the grammar definition should not be part of the data or manuscript, but it should be less prioritized. If the authors are able to provide quality and uncertainty measures of the data, I suggest major revisions. Otherwise, although I think such a large scale soil hydraulic dataset is very valuable and I acknowledge the amount of work, the manuscript should be rejected since the quality cannot be guaranteed.

Specific comments

Page 2 line 12-15, 27-28: Please explain the datasets at least a little if you mention them (e.g. what is SOTER or WISE?) Page 3 line 6-8: If you cannot proof it, please delete this sentence. Page 3 line 23-page 4 line 7: Out of context. Please give some introduction and change the structure. Page 9 line 3: What is the second source? SoilGrids250m is just one. Page 11 line 9-10: Please provide a reference for this calculation (SOC = SOM / 1.724). Where does the 1.724 come from? Page 12 line 24: Add reference for the permeability classes. Figure 1 in the lower blue box: "wilting point" not "witing point" Data file "texture_error_lookup.xls": In row 13 (index 11) the erroneous item is "=50/LS2". Is this correct? Because it is displayed as a "#DIV/0!" in Excel.

Data structure

I suggest to reorganize the structure of your data in the repository to make it more structured:

- the main derived map (.shp or other format)

- metadata (e.g. EstSoil-EH_v1.0_attribute_fields)

- folder with figures

[Figure]

- folder that contains all other information used to derive this map (e.g. SOC rf Model;original estionian soilmap, texture errors, rosetta outputs etc.)

- README
* * *

---

## Referee Comment (RC2) · Anonymous Referee #2 · 21 Dec 2019

Summary

In the manuscript by Kmoch et al. a new countrywide soil dataset for Estonia at 1:10000 scale is presented. Those soil properties are provided which are the most frequently required soil input variables for eco-hydrological modelling, focusing on providing soil data for the SWAT model. The data originates from the Soil Map of Estonia vector dataset (1:10000), which includes information on soil types according to Estonian soil classification, soil quality, number and depth of soil layers, information on course fragments and Estonian texture classes. Numerical soil properties are derived or through using characteristic values of certain soil groups or computing them from available in-

formation, or if data is not available for calculation, data of external dataset is used.

General comments

The scale of the presented soil dataset is outstanding. Detailed information about coarse fragments is unique. Descriptive or categorical type information originating from soil survey is very valuable even if uncertainty is generated when those are converted into quantitative data. The manuscript presents method to derive input information from soil survey data for those models, which require quantitative information about soil properties. This kind of data transformation has several difficulties which authors had to face. Significant amount of work has been put into the construction of the presented dataset, which has to be acknowledged. The work deserves to be published after major revision. Please find hereinafter suggestions for consideration.

Terminology used in international literature should be adapted in the manuscript.

It is not clear what authors mean by "complex text codes" in the abstract.

Please provide more precise information about the meaning of "soil profiles (e.g., layers, depths)" "layer information", which is mentioned in the abstract and introduction.

Under materials and methods section authors mention that potential fertility was mapped, in the abstract and introduction soil quality is mentioned. It has to be clarified which soil property with which method was mapped, and reference or detailed description on how it was derived is needed. A table including metadata would be very informative in the manuscript, in which variable name, file name, description of variable, units of measure, reference, etc. could be included, e.g. meta file of SoilGrids. The "EstSoil-EH_v1.0_attribute_fields.txt" file could be a starting point for that.

The authors could put into context the novelty of providing data at 1:10000 scale – which scale is outstanding. Information on other national soil datasets – which are considered detailed or high-resolution e.g. https://dl.sciencesocieties.org/publications/sssaj/pdfs/82/1/186, etc. – could be

referenced, and the progress presented by EstSoil-EH v1.0 could be highlighted.

Regarding the mapped soil properties, the following specific comments could be considered for the manuscript:

1. Soil type: Is it not clear why new soil types were added to the original dataset, how soil type was extended, e.g. was original soil type recoded based on soil profile information included in the dataset? How were Estonian soil types translated into WRB reference groups? Is there a reference document for it? Based on which soil classification system did you add new soil types and how? Please write down how many soil types were included initially and how many soil types were added. It is not clear how you got 7067 soil types in the attribute table if 120 soil types exist in Estonia. Maybe you meant something different. P4 L28: why "Overall soil type group" is differentiated from "Soil type" which is in L20?

2. Texture classes: Clarification is needed on how USDA soil textural classes and then sand, silt and clay content were derived. Based on present manuscript Estonian soil textural classes were available from the official 1:10000 scale National Soil Map of Estonia. Estonian soil texture class names were translated using USDA terminology. Based on the Estonian texture class names average sand, silt and clay content were added to each soil layers. Please consider to add USDA texture class names based on the average sand, silt and clay content which characterize the Estonian texture classes. Please provide reference for the definition of the Estonian texture classes.

3. Coarse fragments content: It is not clear how - "skeleton indicator number" was derived from the shape and size of the stones and - "inferred rock content (% of volume)" was derived from "skeleton indicator number".

4. Soil organic carbon content: It has to be described why measured SOC data was averaged by soil units in the training dataset for deriving SOC prediction. Was not it possible to use soil profile data to derive the prediction? Predictors used in the random forest method could be listed under materials and methods section. Performance of
SOC prediction could be included in a table. Variable importance could be shown in a figure.

5. Bulk density: It is mentioned that BD is calculated based on texture and SOM, but texture is not included in Equation 4. It has to be considered that moist bulk density is required for SWAT.

6. Potential fertility: It is listed under materials and methods, but not included under the results. Reference or description for the computation would be needed.

7. Organic horizon thickness: Similarly to potential fertility, it is mentioned under materials and methods, but not discussed in results section. Do you mean thickness of A horizon or thickness of soil horizon with accumulation of humified organic matter? Please add reference.

8. Please clearly state for which soil properties the performance could not be analysed because of lacking measured data.

Some parts of the manuscript could be simplified by decreasing mainly technical descriptions, e.g.: P6 L3-P7 L3.

More descriptive plots and tables could be provided for the readers about the derived dataset.

Language revision of the manuscript would improve its readability.

---

## Author Comment (AC1) · 23 Jan 2020

Response to the reviewers' comments

on 'EstSoil-EH v1.0 An eco-hydrological modelling parameters dataset derived from the Soil Map of Estonia', posted to the Interactive discussion. RC1: 'Referee comment', Anonymous Referee #1, 26 Nov 2019 RC2: 'Referee Comment', Anonymous Referee #2, 21 Dec 2019 We thank the reviewers for their valuable comments. We will address the concerns raised by the reviewers point by point below. The response to the Referees are structured in the suggested sequence: (1) comments from Referees RC1/RC2, (2) author's response A: and author's changes in manuscript.

[Figure]

Anonymous Referee #1

(1) RC1: The manuscript of Kmoch et al. describes a methodology for deriving high resolution 3D soil property data of Estonia, which is published with the manuscript. The data basis for the methodology is the National Soil Map of Estonia and the soil properties are derived with a special focus on the parameters necessary for running the SWAT model. These parameters include the saturated hydraulic conductivity, field capacity, wilting point and the USLE K erodibility factor. Such large-scale soil data is highly valuable for soil hydrological and water quality modeling on a scale relevant for decision makers (e.g., national scale). Organizing, homogenization and distribution of soil properties on such a large scale is very challenging and I acknowledge the work the authors did here. However, there are some points that prevent the manuscript from being published ati the current state.

(2) A: We appreciate the initial assessment and thank the reviewer for the value comments. We will address the concerns raised by the reviewer point by point below.

(1) RC1: General comments 1. Structure: The manuscript is very technical and includes too many details. The main step is the transformation of a text based soil classification (Soil Map of Estonia) to soil texture, which is then (together with SOC, bulk density and topographic information) used for deriving the soil hydraulic properties. All the details (especially the grammar definition parts) makes it difficult to follow these main steps. The explanation of all codes for transforming the letter codes to texture could e.g. be a part of the dataset itself as a documentation.

(2) A: We agree, that we have included a lot of very technical information. We believe, that for the soil community in Estonia, and for countries which have similar national datasets like the original soil map of Estonia, these technical descriptions are very interesting and a topic for deeper discussion and evaluation as well. With that said, it is important to acknowledge and emphasize, that the source dataset – the original soil map of Estonia- as described in the article, is not based on modelled but on fully

observed data (e.g. texture, soil profile depth, rockiness, presence of organic layer etc). Systematic mapping of Estonian soils to produce soil map in scale 1:5 000 and 1:10 000 was started in 1954 (Reintam, L., Rooma, I., Kull, A. & Kõlli, R. 2005. Soil information and its application in Estonia. In: European Soil Bureau. Research report. 9, 121-132), with most intensive field studies in period 1965-1969. Generally field mapping was carried out in scale 1:10 000 but in hilly or undulating areas with higher soil diversity in scale 1:5000. In 1982-1988 older mapping data was updated and new areas were included with full-area soil quality (primarily fertility, rockiness, water regime, texture, erodability) assessment. In 1988-1990 soil field studies were performed in non-arable land and new mapping of ameliorated land. Forest soils were mapped in period 1976-1989. During large-scale field mapping of soils the texture was determined in situ based on organoleptic methods and for reference profiles laboratory analyses were performed. This enabled calibration between texture defined by organoleptic method by each researcher participating in field survey and texture determined in laboratory (Estonian Land Board, Explanation to the soil map, https://geoportaal.maaamet.ee/docs/muld/mullakaardi_seletuskiri.pdf?t=20091211092214). As a result of large-scale soil mapping, 119 soil varieties in Estonian national classification system have been distinguished and more than 500 combinations of textural status have been described. About 10,000 profiles (1 profile per 330ha) have been sampled and analysed for characterisation of mineral soils (Reintam, L., Rooma, I., Kull, A. & Kõlli, R. 2005. Soil information and its application in Estonia. In: European Soil Bureau. Research report. 9, 121-132; Reintam, Loit; Kull, Ain; Palang, Hannes; Rooma, Igna (2003). Large-Scale Soil Maps and a Supplementary Database for Land Use Planning in Estonia. Journal of Plant Nutrition and Soil Science-Zeitschrift Fur Pflanzenernahrung Und Bodenkunde, 166 (2), 225−231.). Thus, the texture codes and soil types assigned to the ca. 750000 mapped soil units (polygons) are based on many decades of in-situ land surveying practices and describe quite literally the physical state of the soil based on in-situ assessment. Now, in the article and new dataset, the quality of the extraction and derivation process of the initial texture values

is obviously a very critical step for the whole dataset. In many cases, scientists in Estonia have used their own "scripts" to get some numerical data out that they need for their study areas, however, no standardised approach as presented in our paper is currently available. (3) We added this clarification to the article in section 2.1, However, we came to the decision that reducing the section 2.2.2 of preprocessing the texture codes, where the bulk of the already condensed technical description of the grammar is contained, would not improve overall readability in contrast to one of the original intentions of this dataset – to demonstrate the creation of a numerical dataset from an existing observed dataset.

(1) RC1: 2. Texture: The step of transforming a text coded soil classification into a numerical texture value is a very crucial point of the methodology. All the focus on the "grammar definition" hides the main step of the transformation, which is done with Table 2. However, it is nowhere cited or mentioned how this table was derived. Is it based on the literature or on own data? This table is the main factor influencing your final results, hence it should be carefully described how you come up with this values. Furthermore, this rises the general point of the missing validation of your final soil texture product (that then influences the hydraulic properties). You only validate your grammar-generated codes and do an "expert check". However, the texture itself is not validated with measured data (as far as I understand). You mention on page 9 line 3-9 that you validated it against SoilGrids250m, but it is important to show this validation. An expert check alone is not enough, since other user of your data cannot assess the uncertainty. You need a reliable texture database for validating your results and hence Table 2 (and Table 3). For your SOC prediction you show such a validation and you correctly mention the relevance of validation of your other data on page 15 line 27-29.

(2) A: We acknowledge the case presented here by the reviewer. As there are several arguments outlined by the reviewer we respond line by line:

Ll1-3: We partially agree with the reviewer that the transformation step from the extracted Estonian texture codes to the assignment of numerical values for the fine texture fractions was not well explained, however assigning the numerical values of the sand, silt and clay fractions per texture class is based on Estonian soil experts' knowledge. The emphasis on the grammar was intended to make clear that we had to put more work in to actually retrieve these codes in the first place. We added a description to make it more explicit.

Ll4-6: Lines 5-9 on page 8 are intended to describe that Estonian soil experts (co-authors Arno Kanal & Alar Astover, from the two main natural sciences universities in Estonia which are actively doing research and teaching in soil sciences) technically can assign these fractions to the historically recorded Estonian texture codes. And here again, the emphasis is on the fact, that the original soil map of Estonia is representing observed data. The recorded texture codes represent the in-situ assessment of the national soil mapping and surveying efforts. Explaining the Estonian texture code system would include a lot of technicalities and Estonian language terms, which was not desired. However, we acknowledge that this was not as clearly stated as it could have been. We improved the clarity on the creation of the table.

Ll7-15: "the texture itself is not validated with measured data (as far as I understand)" - We believe, the concern raised here again touches the understanding that the original soil map of Estonia is indeed observed data. But that is why we have to be more clear on how well we extracted the existing texture information from the original soil map. Therefore, we found it important to validate the textual extraction and not so much the decision for the assignment of fraction values for each code. Thus, we described it as our best efforts. In addition, there are detailed studies on reference soil profiles in Estonia, Latvia and Lithuania that relate original soil texture, so called Katchinsky texture system (Kachinsky NA. 1965. Fizika potchv [Soil physics], Vol. 1. Moscow: Moscow University Press[in Russian].) to USDA soil system (Calhoun, T.E., Ellermäe, O., Kõlli, R., Lemetti, I., Penu, P. & Smith, C.W., 1998. Benchmark Soils of Estonia Researched thru Baltic –American Collaboration. Problems of Estonian Soil Classification. Transactions of Estonian Agricultural University, 198, 76-114) and erosion

modelling case studies where based on laboratory analyses transfer functions from Katchinsky to USDA texture classes were developed (Laas, A. & Kull, A. 2003, Application of GIS for soil erosion and nutrient loss modelling in a small river catchment. In: E. Beriatos, C.A. Brebbia, H. Coccossis, A.G. Kungolos (Eds.). Sustainable Planning and Development (525−534).. Southampton, Boston: Wessex Institute of Techonology Press.). Relationship between Katchinsky and Atterberg systems are provided by R. Kask (Kask, R. 2001. On the English Equivalents of the Estonian Terms for the Textural Classes of Estonian Soils. Journal of Agricultural Science, Vol. 14, 93-96. http://agrt.emu.ee/pdf/proceedings/toim_2001_14_kaskr.pdf) But we acknowledge the reviewers desire for numerical statistical hard data which could reveal how well and accurately the soil surveying has captured soil texture in these descriptive codes through the times. But we think, this would be beyond the scope of the current dataset because it would require inclusion of additional independent georeferenced datasets with USDA soil texture classes defined based on laboratory analysis of particle size distribution.

We added additional explanations and background to section 2.2.4.

(1) RC1: 3. Data quality: A data paper should be supportive for the dataset and help the users to evaluate the data and its quality (e.g. uncertainties). This is missing at the moment and instead of focusing on the grammar methodology you should rather present your final derived hydraulic data with e.g. appropriate diagrams.

(2) A: We agree, the data quality is not consistently reported. The texture data and soil types are observed data of many years of national surveying activities. The original observations were classified into the Estonian texture code system based on Katchinsky (1965) soil particle size standards at the time of observation (not by us). We "just" translate" the texture codes back into numbers. Therefore, we don't see a possibility to explicitly display uncertainties related to that process, as we take these as observed data and thus we can make only short reference to another study what shows that achieving 5% accuracy in organoleptic determination of clay content for lower value classes while possible error increased in case of heavy texture classes (Kokk, R.

1987.Soil texture of Estonian soils, its determination and classification. Estonian soils in figures VI. In Estonian: Eesti muldade lõimis, selle määramine ja klassifitseerimine. Eesti mullastik arvudes VI). However, we will compare the sand, silt, clay and coarse fragments fractions with other soil datasets and report on standard deviations and R2. The uncertainties for the SOC predictions are reported. The uncertainties of BD are directly related to the SOC uncertainty. AWC is directly derived from Toth et al. 2017 EU-HydroSoilGrids and we will report the cumulative uncertainty based on our aggregation. K is predicted by ROSETTA based on our derived numerical texture values. We will report the uncertainty from these predictions.

(1) RC1: This includes the uncertainties derived from the texture + SOC classification and also the uncertainty introduced by the pedotransfer functions you used (here ROSETTA). This is also relevant for me as a referee. At the moment for me it is really difficult to evaluate your data in a feasible time. You also mention the problems to derive USDA texture from the old soviet-era based texture system, which ignores the silt fraction and has a different definition for the gravel-sand boundary (page 7 line 29-31). This of course includes a lot of uncertainty, but I understand the benefits of transforming the texture to the often used USDA classification (e.g. usability of pedotransfer functions). I suggest to also include the soviet texture into your data. This can help to evaluate the error introduced by the two different systems and potentially allows to use the data with another "texture transfer function" (different from Table 2).

(2) A: There is no error to be assessed from the translation from one system into the USDA system. The numerical values for sand, silt and clay fractions were assigned to the Estonian texture system codes, then the values were also used to select the appropriate USDA classes based on the texture triangle. .). Relationship between Katchinsky and Atterberg systems are provided by R. Kask (Kask, R. 2001. On the English Equivalents of the Estonian Terms for the Textural Classes of Estonian Soils. Journal of Agricultural Science, Vol. 14, 93-96. http://agrt.emu.ee/pdf/proceedings/toim_2001_14_kaskr.pdf). It is not possible retrospectively to redefine minor differences in boundaries between different classes between texture systems, but we consider natural variation of texture within the soil mapping unit in scale 1:10 000 more significant than that of different texture systems. Texture transfer rules (Table 2 in manuscript) to get from Estonian texture classes to USDA particle size distributions were composed by authors according to Estonian guideline "Field Soil Survey – Muldade väliuurimine" (Astover et al. 2013) where matches of Estonian/Soviet and USDA/FAO classes for field survey is provided. In our opinion it is appropriate approach for data conversion in used mapping scale. We agree that it might increase uncertainty for point data but should be not major problem in case polygon data.

4) Dataset check: By checking randomly sampled polygons in the final GIS product (.shp) I recognized some problems with the soil layers. E.g. FID 96775 has two layers with SOC and bulk density values are shown in layer 1 and 2. However, texture values are indicated in layer 1 and 3, whereas layer 2 is empty. Similar problem was found in FID 178514 with only one layer but texture values in layer 1 and 2. Please check your data again.

(2) A: Thank you for pointing these errors out. We believe they were introduced when making assumptions about layers that have no depth reported, but are above or below a layer that has a depth reported. We are uploading an updated dataset.

(1) RC1: 5. SWAT focus: The manuscript focuses too strongly on SWAT. Although the dataset was created for using it with the SWAT model, this is not important in the data paper. Of course you can mention that the presented data is enough for many modeling purposes (e.g. SWAT), but at the moment the focus on SWAT makes the manuscript difficult to understand. E.g. on page 7 line 6-15, just mention that you have defined different layers.

(2) A: We acknowledge the reviewer's suggestion, we generalised to the need for eco-hydrological modelling, and reduced the focus on SWAT throughout the manuscript.

(1) RC1: 6. Highlight the need for your dataset: You mention similar global or regional datasets (page 2 line 4-28). However you miss to highlight the need of your dataset. What is different from the others or "better" in your dataset? Why it needs a new dataset? For calculating the available water capacity you use the dataset of Tóth et al. (2017) which is not mentioned in this section. Why is this dataset not usable for parameterizing models in Estonia?

(2) A: Other datasets are available for use in Estonia-based modelling contexts, that is correct. However, vector-based EU soil datasets are very coarse and excessively generalise large parts of the diverse Estonian landscape. High-detail datasets such as Soilgrids are themselves based predicted on a grid 1km/250m, and not based on observed data. The presented dataset is of very high spatial detail based on the original Estonian national soil map, which was created from directly surveying all of Estonia. Thus, our presented dataset much more spatially related to the landform/landuse observed there. Furthermore, the textures and SOC/BD values are directly derived from reliable observed data samples from Estonia, with a reproducible workflow, whereas this is not true for many other reported soil dataset that covers the area of Estonia. Furthermore, the method created to translate original hard copy soil map (with traditional textual codes) to digitally readable GIS-based map can be used by several other countries (e.g. Latvia, Lithuania, Ukraine etc) and this enables spatially more explicit modelling of ecosystems.

We added the explanation as highlights to the manuscript in the introduction.

(1) RC1: In summary the manuscript should rather focus on the quality of the data than on the methodology of the grammar definition. That does not mean that the grammar definition should not be part of the data or manuscript, but it should be less prioritized. If the authors are able to provide quality and uncertainty measures of the data, I suggest major revisions. Otherwise, although I think such a large scale soil hydraulic dataset is very valuable and I acknowledge the amount of work, the manuscript should be rejected since the quality cannot be guaranteed.

(2) A: We thank the reviewer for the valuable comments. However, we want to turn the interest of the reader also to one of our original motivations, which was to provide methodological approach of getting from legacy qualitative soil map to quantitative functional maps.

(1) RC1: Specific comments:

(1) RC1: Page 2 line 12-15, 27-28: Please explain the datasets at least a little if you mention them (e.g. what is SOTER or WISE?) (2) A: added to the manuscript

(1) RC1: Page 3 line 6-8: If you cannot proof it, please delete this sentence.

(2) A: deleted.

(1) RC1: Page 3 line 23-page 4 line 7: Out of context. Please give some introduction and change the structure.

(2) A: added to the manuscript

(1) RC1: Page 9 line 3: What is the second source? SoilGrids250m is just one.

(2) A: The two sources were the manually decoded dataset from the paragraph before and the second was SoilGrids250m. We rephrased.

(1) RC1: Page 11 line 9-10: Please provide a reference for this calculation (SOC = SOM / 1.724). Where does the 1.724 come from?

(2) A: Coversion factor 1.72 is widely used universal value (Soil Analysis: An Interpretation Manual. Eds. K. I. Peverill, L. A. Sparrow, D. J. Reuter. CSIRO Publishing, 2001; Soil Carbon Dynamics: An Integrated Methodology. Eds. W. L. Kutsch, M. Bahn, A. Heinemeyer. Cambrige. 2012), however we acknowledge that the real value varies slightly between soils.

(1) RC1: Page 12 line 24: Add reference for the permeability classes. Figure 1 in the lower blue box: "wilting point" not "witing point"
(2) A: we updated the figure and added the reference

(1) RC1: Data file "texture_error_lookup.xls": In row 13 (index 11) the (1) RC1: erroneous item is "=50/LS2". Is this correct? Because it is displayed as a "#DIV/0!" in Excel.

(2) A: it is not erroneous. The cell code is "=50/ls", which is an invalid texture code. Excel interpretes the "="

(1) RC1: Data structure: I suggest to reorganize the structure of your data in the repository to make it more structured:

- the main derived map (.shp or other format) - metadata (e.g. EstSoil-EH_v1.0_attribute_fields) - folder with figures - folder that contains all other information used to derive this map (e.g. SOC rf Model;original estionian soilmap, texture errors, rosetta outputs etc.) - README

Anonymous Referee #2 Summary

(1) RC2: In the manuscript by Kmoch et al. a new countrywide soil dataset for Estonia at 1:10000 scale is presented. Those soil properties are provided which are the most frequently required soil input variables for eco-hydrological modelling, focusing on providing soil data for the SWAT model. The data originates from the Soil Map of Estonia vector dataset (1:10000), which includes information on soil types according to Estonian soil classification, soil quality, number and depth of soil layers, information on course fragments and Estonian texture classes. Numerical soil properties are derived or through using characteristic values of certain soil groups or computing them from available information, or if data is not available for calculation, data of external dataset is used.

General comments

The scale of the presented soil dataset is outstanding. Detailed information about coarse fragments is unique. Descriptive or categorical type information originating from

soil survey is very valuable even if uncertainty is generated when those are converted into quantitative data. The manuscript presents method to derive input information from soil survey data for those models, which require quantitative information about soil properties. This kind of data transformation has several difficulties which authors had to face. Significant amount of work has been put into the construction of the presented dataset, which has to be acknowledged. The work deserves to be published after major revision. Please find hereinafter suggestions for consideration.

(2) A: We appreciate the initial assessment and thank the reviewer for the value comments. We will address the concerns raised by the reviewer point by point below.

(1) RC2: Terminology used in international literature should be adapted in the manuscript. It is not clear what authors mean by "complex text codes" in the abstract.

(2) A: The Estonian texture information field in the original soilmap's attribute table is comprising not only of one actual texture class, but joined with classifiers for the rock content, peat soils and distinct compositional layers and their depth. Visual examples of the meaning "complex text code" of the soil map are shown by Reintam, L., Rooma, I., Kull, A. & Kõlli, R. 2005. Soil information and its application in Estonia. In: European Soil Bureau. Research report. 9, 121-132 and Reintam, Loit; Kull, Ain; Palang, Hannes; Rooma, Igna (2003). Large-Scale Soil Maps and a Supplementary Database for Land Use Planning in Estonia. Journal of Plant Nutrition and Soil Science-Zeitschrift Fur Pflanzenernahrung Und Bodenkunde, 166 (2), 225−231.) and as an example of texture code of Skeletic Leptosol is shown "ls_110-20/pk;r_4ls_1" which indicates presence of sandy silt loam, layer depth, solid limestone bedrock, rock content 50-7%. We have included more information in the manuscript regarding the texture attribute field in the abstract, and sections 2.2.2 and 2.2.4.

(1) RC2: Please provide more precise information about the meaning of "soil profiles (e.g., layers, depths)" "layer information", which is mentioned in the abstract and introduction. Under materials and methods section authors mention that potential fertility
was mapped, in the abstract and introduction soil quality is mentioned. It has to be clarified which soil property with which method was mapped, and reference or detailed description on how it was derived is needed.

(2) A: The field surveys for each polygon were made by sampling the soil to a default depth of one metre and the describing the visual and organoleptically defined texture of the soil from the surface to the depth of one metre. This is considered the observed soil profile. The assessed information was then noted on the field data sheet.

We added more information to the section 2.1.

(1) RC2: A table including metadata would be very informative in the manuscript, in which variable name, file name, description of variable, units of measure, reference, etc. could be included, e.g. meta file of SoilGrids. The "EstSoil-EH_v1.0_attribute_fields.txt" file could be a starting point for that.

(2) A: We thank the reviewer for the suggestion and will add an overview table for the derived variables.

(1) RC2: The authors could put into context the novelty of providing data at 1:10000 scale – which scale is outstanding. Information on other national soil datasets – which are considered detailed or high-resolution e.g. https://dl.sciencesocieties.org/publications/sssaj/pdfs/82/1/186, etc. – could be referenced, and the progress presented by EstSoil-EH v1.0 could be highlighted.

(2) A: We added the reference to the introduction in order to highlight how the EstSoil-EH v1.0 dataset preparation relates.

(1) RC2: Regarding the mapped soil properties, the following specific comments could be considered for the manuscript: 1. Soil type: Is it not clear why new soil types were added to the original dataset, how soil type was extended, e.g. was original soil type recoded based on soil profile information included in the dataset? How were Estonian soil types translated into WRB reference groups? Is there a reference document for it?

Based on which soil classification system did you add new soil types and how? Please write down how many soil types were included initially and how many soil types were added. It is not clear how you got 7067 soil types in the attribute table if 120 soil types exist in Estonia. Maybe you meant something different.

(2) A: The Estonian national soil data set describes a base set of soil types, unfortunately in different literature 120-130. The difference is mainly caused because some researchers add to soil types also non-soil surface types (e.g. soilless bedrock; in total 7 types) and/or distinguish some subtypes of main soil types (e.g. Krf – recultivated open pit mining soil). It describes the most generalized level of Estonian soil classification for mapping that can be extended from these main 120-130 classes, incl. level of erosion, slope position, similar to FAO secondary identifiers, but instead of keeping them separate they just extended the main class, as the inherent "grammar" is well known in Estonia. This increased the overall number of explicit soil classes to more than 7000. We "just" extracted the main soil classes again.

(1) RC2: P4 L28: why "Overall soil type group" is differentiated from "Soil type" which is in L20?

(2) A: Original soil type is main category in Estonian soil classification and soil type group includes several soil types with main similar features. We clarified with more information in sections 2.1 and 2.2.1.

(1) RC2: 2. Texture classes: Clarification is needed on how USDA soil textural classes and then sand, silt and clay content were derived. Based on present manuscript Estonian soil textural classes were available from the official 1:10000 scale National Soil Map of Estonia. Estonian soil texture class names were translated using USDA terminology. Based on the Estonian texture class names average sand, silt and clay content were added to each soil layers. Please consider to add USDA texture class names based on the average sand, silt and clay content which characterize the Estonian texture classes. Please provide reference for the definition of the Estonian texture

classes.

(2) A: The process followed a different order: The Estonian texture classes (Katchinsky, 1965 system), based on their known composition (based on how they were encoded at the time of survey) have numerical values for sand, silt, clay fractions assigned. Then, based on these numbers the USDA texture names were assigned. The single important reference is table 2 that designates the sand silt and clay fractions for each known Estonian texture types

(1) RC2: 3. Coarse fragments content: It is not clear how - "skeleton indicator number" was derived from the shape and size of the stones and - "inferred rock content (% of volume)" was derived from "skeleton indicator number".

(2) A: We acknowledge that this was not well explained. Similar to the Estonian texture classes there exist Estonian stoniness classes, that describe a certain type of coarse fragments within the soil profile. An additional number in connection with this rock type identifier indicates the amount/volume of these rocks in 1kg of soil. We used this to characterise the coarse fragments. We clarified this in section 2.2.4

(1) RC2: 4. Soil organic carbon content: It has to be described why measured SOC data was averaged by soil units in the training dataset for deriving SOC prediction. Was not it possible to use soil profile data to derive the prediction? Predictors used in the random forest method could be listed under materials and methods section. Performance of SOC prediction could be included in a table. Variable importance could be shown in a figure. (2) A: As several of the Estonian soil units (which contain the predictor soil profile information) would have contained a significant amount of sampling points for SOC (basically a few experimental trenches on agricultural fields) all the predictor variables (including the soil type, texture and topographical variables) aggregated for these units would be identical, this would have created a very strong bias in the training of the model. Therefore, SOC data points that lie within the same soil unit were averaged.

(1) RC2: 5. Bulk density: It is mentioned that BD is calculated based on texture and SOM, but texture is not included in Equation 4. It has to be considered that moist bulk density is required for SWAT.

(2) A: Depending on the formula indeed we are not using texture at all. Our real SOC/SOM field measurements and thus the modelled values are for dry bulk density. Removing the focus on SWAT as suggested by Reviewer1 would mitigate the conflict to provide moist bulk density. We state that this is dry BD, based on the dried SOM measurements.

(1) RC2: 6. Potential fertility: It is listed under materials and methods, but not included under the results. Reference or description for the computation would be needed.

(2) A: It was a historical data field in the original soil map. We did not use this field as the reliability and its original calculation is questionable. We remove notion of this field.

(1) RC2: 7. Organic horizon thickness: Similarly to potential fertility, it is mentioned under materials and methods, but not discussed in results section. Do you mean thickness of A horizon or thickness of soil horizon with accumulation of humified organic matter? Please add reference.

(2) A: See above, that is correct. For now we are not considering this field. Possibly we should? It could improve some data regarding organic matter/SOC? However, we are not decoding it. Same for the rockiness ('kivisus')

(1) RC2: 8. Please clearly state for which soil properties the performance could not be analysed because of lacking measured data.

(2) A: Yes, we have to be more precise/consistent about this. We added the reporting of data quality to manuscript.

(1) RC2: Some parts of the manuscript could be simplified by decreasing mainly technical descriptions, e.g.: P6 L3-P7 L3.

(2) A: We partially agree with RC2. However, we deem these technical details are already greatly reduced.

(1) RC2: More descriptive plots and tables could be provided for the readers about the derived dataset.

(2) A: Reviewer 1 also suggested plots regarding uncertainties and data quality. We will add these to the manuscript.

(1) RC2: Language revision of the manuscript would improve its readability. (2) A: The manuscript had already been edited by English native professional scientific editor.

---

## Referee Report (RR1)

[referee-annotated manuscript omitted]

---

## Author Response (AR2)

**Response to review comments**

Reviewer #1:

The authors present a national soil data base for Estonia. The dataset comprises data from different sources and focusses on an application in SWAT models. The manuscript describes the many challenges which needed to be resolved on the way to derive this very valuable resulting data base. The core novelty of the manuscript appears to be i) the derivation of standardised data for SWAT modelling at the scale and scope at hand. This has been achieved in four steps a) the remapping of the different soil taxonomies, b) derivation of topography-based site characteristics, c) application of a random forest to SOC and BD extrapolation, and d) an application of Rosetta and estimation of the USLE k parameter.

The manuscript improved from the initial version. The dataset is highly valuable and deserves publication. However, the authors provide a plethora of information, which is very difficult to connect and trace through the manuscript. Especially when "expert knowledge" converts to numerics it remains challenging to understand the steps and the data. For each step the authors apply a set of concepts. But they do not really clarify on these concepts and about the respective implications for the usage of their data product.

I have full confidence that the authors can and will improve these shortcomings and hope that the following comments can assist their revisions.

We appreciate the valuable comments and hope to have addressed the points raised.

General Comments

As a data paper there appears to be a structural challenge with the manuscript. On the one hand it is important to report the resulting data, including transparency about the sources and validity of the derived values. On the other hand the methods and challenges to derive this data are taking a lot of room, which obscure clarity of the presentation of the methods and results. I suggest to rigorously revise the structure and content of the manuscript under the view of "how to provide the readers the means to understand and work with the dataset".

We have restructured the document with attention on the workflow steps as depicted in the main figure 1. We have reduced the technicalities a lot and put them in an additional technical supplement. In the results section we have added additional error metrics and characteristics for the reader to evaluate various use case, e.g. in arable lands, grassland or forested areas.

I am sure the authors aim to do so from the start. Figure 1 is capable to provide a solid basis for this. Hence, the introduction could work exactly towards the four major steps/aspects plus a brief clarification of the motivation of the study. In the present state, my capabilities to distill information from the introduction is massively exceeded. One possible avenue to follow could be:

Analysing effects of land management decisions is an important issue. SWAT-type models are used and heavily rely on soil data at high resolution and level of detail. Although a global issue, it is found at National level as well. This study is about Estonia where - as in many other places - mapped soil data is very difficult to convert to numerical data for the intended model application. Moreover, the required aspects/variables are lacking in the original soil maps. This study presents a soil data base, which includes layering, texture and pedo-physical variables as well as site characteristics…

We have restructured the manuscript added additional background for several paragraphs where the expert inputs came in, which now also underpinned with references like an EU project report where several decisions have been made.

Further, I am under the impression that some of the debate (i.e. about SOC) might rather suit the discussion section?

We have reorganised some sections in the introduction and moved a paragraph also to the discussion section.

The methods-section gives a lot of information. Although I find it penetrable after some intense study, I feel somewhat overwhelmed and left alone at the same time. I think the structure is already laid out well in Figure 1 and requires to be strictly followed. Hence I suggest to offensively start with Figure 1 and motivate the different aspects as classical soil data, topographic data, specific soil variables (SOC, BD), pedo-hydrological parameters. This methodological introduction could include the general idea behind each step. The remainder of the section can host the respective methodological steps. I would suggest to give each subsection a similar pattern: Available data, required data, how to come from a to b, how can we evaluate the method, where is the code. Through this, I expect that it becomes much easier to follow, the subsections will become shorter and less technical. Moreover, I sincerely urge the authors to point out the conceptual ideas instead of listing the parsing grammar and error handling. I agree that this is important, but at the current state I feel massively distracted by reading strings which parse and links to external files.

We have reduced the technicalities a lot and put them in an additional technical supplement.

The tables 1 and 2 might suffice to document the remapping with a little explanation?

Yes, that is right and was our intention from the get-go. We added also a technical report link to back up choices of "expert knowledge".

Your second step distills topography-related site properties (Sec. 2.3). I am sure that not only an estimate of SOC concerns with such information. E.g. HAND and similar indices (Gharari et al. 2011, Loritz et al. 2018) are pointing out the use of such information. Maybe a simple table with the applied indices can be listed and that is enough?

We appreciate the provided information. The terrain related indices we only used for SOC prediction under the hypothesis that slope affects deposition and thus erosion and mineralisation are indirectly correlating to the level of SOC. However, clay and sand content, soil group and land use were by far the strongest indicators and only then came additional terrain indices with very low influence. A new and more complex model could likely be a paper on its own as reviewer 2 implies, too.

The derivation of SOC appears maybe to be the most vague point since it employs few observations and many assumptions. Hence, I would see this step and the subsequent application of Rosetta and calculation of USLE_K to be maybe somewhat a different level of "data" and should be treated as such? I consider it out of the scope of your study to really prove the effect different ways to reproject the soil types and to derive topographic indices on the PTF and RF. In the same pattern as before, I think it can be handled rather straight forward to present the foundation data and the PTF / RF application. Again, I think a table/figure what inputs convert to which outputs with what assumptions would likely elucidate on the step much concisely?

Based on reviewer 2 comments we dropped USLE_K as immediate variable due to weak points in additional variable derivation for USLE K equation. However, hydraulic conductivity is only based on

texture properties sand, silt and clay. We explicitly omit the variation where we could use BD as additional parameter for Rosetta due to the high uncertainty. For the calculation of BD it is expressly stated that it depends only on the SOC step during the same section in the manuscript.

I have seen some passages about possible evaluation strategies of the derived data. Since this is a data publication, I would suggest to give this a much stronger focus. Again, most appears to be there and might require just a little more clarification. It could follow the same pattern in each aspect.

For the validation of SOC I was wondering if the random split sampling results in subsets which still cover most of the sample patches. As such, the validation would maybe not give validation for spatial extrapolation. Could you check to cross validate between different regions? As the data set appears bimodal, I am also not quite convinced about the validity of a global R2 as "uncertainty" estimate. For the cluster of moderate values, there are deviations of >400%! I think this application is a valid approach, however there should be clear notice of the relatively large uncertainty - especially since SWAT modelling is intended as application where people might relate findings to the reported SOC values.

We put a lot of effort into additional error descriptions. We added RMSE and the normalised median absolute deviation error metric to SOC, subsets of samples where observed data for BD is available, as well as sub samples for sand, silt and clay. We refer to the metrics and there limited explainability over the whole dataset, however, we also break down SOC prediction error for different landforms with additional descriptive statistics, such as error quantiles. This will help the user of the data to make choices that are more informed.

The results section should present the derived data and maps in a brief manner. I can imagine that an essence of Fig. 4-9 could be shown here in max. 4 panels. I would expect a description of the dataset here. Referring to Table 6 the different "sections" could be related back to the applied methods. Instead of the technical definition of the data_type I would favour a clarification about the data reliability, which could come in the remainder of the section as evaluation of each processing step. Again, I think most of the material is there and the authors can be very specific about each evaluation finding. In the present form, the results highly underrate the valuable outcome of the study and give little guidance about how to use it. Rather technical details could be given as appendix (e.g. texture remapping) and make room to really point to the reliability of the compiled data.

As explained above, we added additional error metrics, broken down for main land uses in Estonia (arable lands, open grassland, forest and wetlands). We the panels suggestion sounds great, but fear for the lack of quality/visibility if we reduce the image sizes.

As stated before, I would see many paragraphs in the introduction and methods section actually suiting the discussion section. I think the authors could remain more specific to their estimates of step C and D instead of this rather general discussion. This would also be a perfect place to link to the topic of the special issue and highlighting where soil mapping and landscape functioning are challenging/easy to connect.

We also added several sentences to link better to the special issue.

As I suggest to rework the manuscript rather strongly, I omit to give specific comments at this stage. However, I would remove "End" in Fig. 1. Moreover, I would remove most of the file names and links from the text and either put them to the Bibliography or as a file description in the Appendix.

We reworked the manuscript a lot. We also removed many of the links to scripts and supplemental materials.

General comments:

Authors have answered most of the questions raised about the first version of the manuscript. They have improved the manuscript based on the suggestions, but clarity and information on uncertainty needs further input. It is understandable that quantifying the uncertainty of the variables is quite challenging for the authors. It would need significant amount of time to perform it, especially to find the correct method for each mapped variable/soil property. It might be an interesting analysis for a separate paper. The presentation of the present state of the dataset is already valuable. In this manuscript the main strength is the description on how the enormous and very valuable soil data collected in line with the Soviet system could be translated into a dataset which can be used for quantitative analyses. The followings should be further described and highlighted in case of all mapped variables under discussion:

- how the variables were derived, e.g.: based on expert rules/ characteristic mean values are assigned/ computed with PTFs based on characteristic particle size distribution, etc.;

- what are the limitations of their use.

We appreciate the valuable comments and have addressed the points raised. We have added error metrics where observed data was available, even if only for partial subsets of the data. We thus also addressed the limits and preferable use cases, especially for landforms where errors for predictions are higher or lower.

Still terminology used in international literature should be adapted in the manuscript. It is needed not only for the texture, but for all expressions related to soil science in the entire text, tables, e.g.: in WRB the word "soil type" is not used, there are reference soil groups (http://www.fao.org/3/i3794en/I3794en.pdf). Please provide information in the text about how WRB reference groups and qualifiers were derived and mention possible limitation of providing this information.

We have changed the wording across the manuscript where appropriate. After all, the Estonian terminology only calls it soil type. But we provide guidance, from where we transition the user towards the wording of soil reference group. Unfortunately, one important soil expert who provided the intial insights has passed away during the work on the manuscript (Dr. Arno Kanal). In fact, he was one of the few experts who was familiar with the Soviet system. Notwithstanding his tragic loss, we could support and extend his work with collaboration with the Ministry for Environment in Estonia, where we gained access to additional observed data to calculate error statistics for fine earth fractions and bulk density. We also link to a report where Estonia has been collaborating in a large EU project (BioSoils), where WRB soil reference groups where designated for various Estonian soil types.

May be the derived particle size distribution (SOL_SAND, SOL_SILT, SOL_CLAY) could be referred in the manuscript as clay, silt and sand content characteristic for the texture class of the unit and you could add reference of the conversion method.

In the flow text we now only refer to sand, silt and clay content or fractions. Only when referring to the variable name in the dataset we left the SOL_ prefix. The conversion is based on the tables 1 and 2 as explained in the text. We have put more emphasis on this, to be clearer to the reader. The values for sand, silt and clay content have also gained some error metrics, but only for forest area was observed data available.

Further checking the grammar and terminology would be important, e.g. among others P15 L13, P17 L9, L12-13, P18 L15. Some suggestions are provided under specific comments.

On the map (EstSoil-EH_v1.0.shp) available from https://zenodo.org/record/3473290#.XmdtHkq6qUk there is no information about "wrb_main", "EST_CRS1-4", "Huumus", "ao_hor_thick", "ao_hor_type", "geometry" are mentioned in Table 6.The map includes "Boniteet" and "Varv", but not included in Table 6. You mention that you might not include information on soil fertility. Please harmonize Table 6 and the attributes of the map.

We updated Table 6 (now table 5) to focus only on the described parameters in the paper. There are additional variables in the dataset which where temporarily used during the machine-learning process. On request from users we let them in.

Just randomly checking the database for some cases it was not logical that nlayers=1 and clay, silt, sand and rock content is given for layer 2 as well, e.g.: orig_fig=194041, 191116, or nlayer is 2 and clay, silt, sand and rock content is given for layer 1 and 3, e.g.: orig_fig=372656. It might be useful to not use zero (0) for those rows where there is no data. Please check and correct these.

We agree, that instead of zero (0) we should employ NULL or similar measures in order differentiate between a value of 0 and no value. However, the use of the dataset implies to only use the variables which have defined layers. We have uploaded an updated version of the dataset.

Please clarify in the text the followings, which were mentioned in the first round of the review as well:

- How were Estonian soil types translated into WRB reference soil groups and qualifiers? Is there a reference document for it?

Yes, there is a reference report now added. Initially we had only received "expert knowledge" input, but now it is underpinned with a documented EU project (BioSoils).

- Predictors used in the random forest method could be listed under materials and methods section.

Specific comments:

We have significantly restructured and revised the manuscript. We hope to have addressed all specific points.

P2 L9-12: please finish the sentence

Edited.

P2 L14: please reduce repetition of soil and terrain

Removed.

P2 L15: Please add meaning of ESD.

Added.

P2 L20-21: … soil information derived with machine learning methods …

Edited.

P2 L23-24: … sand, silt and clay content, amount of coarse fragments, organic carbon content and carbon stocks at seven soil depths … please use the word "content" in the entire text in case of the above mentioned soil properties.

We have edited the manuscript accordingly.

P2 L25-26: please consider that also SoilGrids provide harmonized soil database for Europe, please rephrase the sentence. The following can be deleted: ", and also covers Estonia", it is logical.

Edited.

P2 L30-31: it is not clear why HYPRES dataset is mentioned. In this case LUCAS or EU-HYDI datasets could be mentioned as well. Please consider the message of the text and revise the sentence accordingly.

We have added EU-HYDI.

P3 L23: … related to water and carbon cycle … is it correct? AWC is missing from the listing.

Yes, we added cycle in several occasions. We added AWC.

P3 L27: … There is no countrywide spatial dataset of soil organic carbon content and bulk density for Estonia ... is it correct this way? Please finish this thread, e.g.: it was needed to derive predictions for both soil properties which made it possible to map them.

We have added the clarification.

P3 L27-30: could be moved after L18.

P4 L18-19: instead of soil profiles would it be appropriate to write soil layering?

Soil profile related also to the sampling process as combined wording for layers and taking of physical sample.

P4 L20: … related to water and carbon cycle… is it correct?

Yes, added.

P4 L21: … from the historical soil maps of Estonia – surveyed between 1949 and 1991 – to support modelling … is it correct?

Indirect. We added clarification that the original soil surveys mainly were done for land evaluation and planning and assessment of agricultural use.

P5 L1: … based on organoleptic field judgement (feel methods) and …

Edited.

P5 L6: … combinations have been described considering the texture of soil layers … is it correct?

That was not clear to us. The sentence does not exist anymore.

P6 L2: the following can be deleted: "instead of 9, 21 (9+12) or 108 (9x12)". It is not clear why 87240 unique values are recorded in the previous dataset. Does it come from the combination of soil type + texture + layering + level of erosion + slope position – similarly to the explanation you provided under previous answers?

These technical descriptions have been moved into a technical supplement. But your assumption is correct.

P6 L6: … to derive …

Edited.

P6 L19: please provide reference literature for translating Estonian soil types to WRB reference soil groups and qualifiers.

Added as mentioned above already.

P8 L27: … to USDA texture classes … is it correct?

Yes.

P9 L13: How did you calculate the accuracy of organoleptic determination of clay content? Through the organoleptic determination didn't you determine the soil texture class? Or do you determine directly the clay content? Please clarify it.

The organoleptic determination resulted only in a categorical label. The translation into a numerical value is based on Table 2 and "expert knowledge". We have now also several numerical error metrics (at least for forest soils).

P10 L12: … We compared the derived sand, silt and clay content values with two different datasets. … is it correct?

Correct, soil grids and a county level size manual based extract.

P10 L22, L26: on P13 L20 you mention that Ksat was calculated with Rosetta PTFs, thus it was not derived from EU-SoilHydroGrids. Please revise it.

That is correct, we have not stated that K sat is derived from EU-SoilHydroGrids, only AWC.

P10 L27-29: Please describe more detailed the differences between EstSoil-Eh and SoilGrids.

We introduce SoilGrids already in the introduction, main difference is possibly that EstSoil is vector based and formed from categorical data and SoilGrids is raster based and formed from nurmerical samples. This is also stated in the discussion.

P12 L15: … as predictor variables for the calculation of SOC and BD …

Edited.

P12 L21: please note that number of randomly selected variables – during each split – and number of trees in the forest are usually optimized.

Ok.

P12 L30: the following can be deleted: "for machine learning".

Deleted.

P13 L3: please list predictor variables.

The technical code supplement "04_soilmap_SOC-RF_BD.ipynb" describes the modelling process in – depth. But the manuscript also states that the models uses the fine earth fractions (sand silt clay content), coarse fragments, topographic variables as in section 2.3 , and drainage where used as predictors.

P13 L5: based on your answer and Equation 4 texture was not used to calculate BD, please delete the following: „texture values and".

Texture values related to sand, silt and clay values, which were used to predict SOC, and SOC was then used as PTF for BD. Deleted.

P13 L6: please shortly describe why you choose that PTF to compute BD, why not other PTF was used, e.g.: applicability/ training set used to derive the PTF was similar.

Added a short comment, that is was already used successfully in Estonia before.

P13 L9: there is no information in the brackets, please check it.

Deleted.

P13 L17: … We included two variables …

P13 L23: … Rosetta3 …

*Edited.*

P14 L10: please check the reference, humic or peaty topsoils do not have blocky, platy or massive structure.

*We removed the USLE K based on your recommendation.*

P14 L11-12: Please provide more information with reference about how soil structural class was derived. Based on solely texture and amount of course material structure cannot be given.

*We removed the USLE K based on your recommendation*

P14 L1-27: based on the present information, deriving structural class is a weak point. Therefore, I would suggest to not include the USLE K erodibility factor in EstSoil-EH dataset and in the manuscript.

*We removed the USLE K based on your recommendation*

P16 L10-11: It is the repetition of the information given in materials and methods, therefore could be deleted.

*deleted*

P16 L25: Is it correct that accuracy of BD could not be analysed because there is no Estonian dataset with measured values? If that is true, please mention it. If there is a dataset where accuracy can be calculated, please perform the analysis.

*Partially correct, we finally obtained some data.*

P16 L27: please describe map of BD as well.

*Figure 6*

P16 L29-30: mentioned in the materials and methods, please delete the sentence.

*Deleted.*

P17 L1-2: if structure class cannot be derived with a more robust method, I would suggest to delete information on USLE K from the entire manuscript and the database.

*We removed the USLE K based on your recommendation*

P18 L14-15: may be the following could be considered: … with a reproducible workflow, which is unique in the case of Estonian soil datasets …

*added*

P17 L24-25: "are informed to some extent by previous reports" it is not clear please rephrase it.

*Rephrased, aiming at SoilGrids that had likely also sample information available from Estonia.*

P17 L30-31: sentence starting with "A direct" is not clear, please rephrase it. It would be better to move the sentence starting with "From the point" under results section.

Edited.

P17 L31-32: … based on the layering of the original texture code per mapped soil units…

Edited.

P18 L8-12: It is a very good idea to use an additional class for peat, but the following sentence might be confusing therefore should be revised or deleted: "From that perspective peat soil units are currently modelled with assumptions to have a similar behaviour to clay hydrologically." Several studies have shown that the shape of the shrinkage characteristics of peat soils were significantly different from those of clay soils (Van den Akker and Hendriks, 1997; Oleszczuk et al., 2003; Hendriks, 2004.)

Much appreciated hint. We will look into it. Currently, seem to have some success with using it in SWAT like this for runoff behaviour.

[revised manuscript text omitted]

---

## Author Response (AR3)

**Topical Editor Decision: Reconsider after major revisions** (24 Aug 2020) by [Conrad Jackisch](Conrad Jackisch)
Comments to the Author:
Dear Alexander and co-workers,

Thank you very much for your intense work on the manuscript. It really gained a lot more conciseness. I am now in the unpleasant situation to ask for another (hopefully largely technical) iteration before I can send it to the reviewers. The reason for this is that I anticipate rather superficial comments since the flow of the manuscript is not yet to the point. I am fully confident that the following suggestions are easy to work through and that it will help the next iteration to finally get the manuscript into a publishable shape. I apologise to cause you work but I am sure it is to the best of your manuscript.

**We appreciate your input and your suggestions have definitely improved the manuscript a lot!**

Abstract:
P1L22-24 (I refer the pages and line numbers to the annotated version in your reply letter) is the most important point here (in my view). The first paragraph appears to be more a relict of the first version. I also do not understand in which way the scales are of importance. Later in the manuscript you relate to SoilGrids as reference but this comparison is not at all summarised in the abstract. Moreover, I was wondering if the title might be amended to somehow relate to the large sources in addition to the "Soviet Soil Map".
What I have in mind is especially that the situation you worked through is not uncommon. There are soil maps in many countries but the eco-hydrological process properties/parameters are very difficult to distill. You provide one example how to do this plus the final data to use SWAT in Estonia. So the central aspect of your manuscript should really emphasise on this as an example.

***Thank you for this recommendation. We took this into consideration and re-organized the manuscript.***

Links as citations:
Although I can understand the notion of using the weblinks in the manuscript, I think the proper way would be to cite the sources similar to paper references. Websites will also require a date for the last access. Please see the guidelines here:
[https://publications.copernicus.org/for_authors/manuscript_preparation.html](https://publications.copernicus.org/for_authors/manuscript_preparation.html)

***We reduced the web links in the ms and formatted them according to the journal guidelines.***

Introduction:
P2L4-6 are central to me. I consider it a better start of the manuscript to state the problem instead of referring to SWAT. This also immediately opens your story: There is detailed soil information but eco-hydrological and pedi-hydrological process modelling (and understanding) requires information on different properties at finer scales. Then you can generalise what information exists in soil maps and global datasets and at what resolution.

***We re-organized and re-wrote the introduction to better address the general problem and introduce the reader to the topic.***

I prefer to give the scale triplet (Western and Blöschl, 1999) instead of a map scale which I have little

idea about the actual data density and validity. At the moment there are 1:10000, 30 arcsec, 250m, 1:500000, 1km etc. given. But I find it very difficult to get an idea about these resources, how the data has been derived and for which scale it might be applicable. I.e. remote sensing of soil properties in temperate climate is not very reliable nor can it give information about the soil profiles over depth. However they provide the advantage of spatially continuous data. You also point to the many soil sampling campaigns leading to the existing soil map. The soil samples are exactly opposite to remote sensing as they provide very detailed information for some points but it remains challenging to transfer this information to a spatially continuous map. I see this in close relation to the overview about data you use as additional inputs and data which is used as references plus the actual challenge of your study. Moreover, this is a fundamental link to our special issue I really think you should exploit for your argumentation.

***Unfortunately, we cannot represent these datasets in scale triplet because all of them are representing continuous data and achieved in either generalisation, interpolation or statistical modelling. The final map scale is chosen based on the input data density (e.g. number of soil profiles per square km) and spatial resolution of environmental covariates that have been used for statistical modelling.***

The introduction features so many possible data sources but it actually gives no outline of your manuscript. This comes P6L10ff. Somehow also 2.1 is hardly discernible from the introduction this way. I propose to restructure here. The introduction needs to guide the reader into your topic. The long list of soil-related resources can be the first subsection in the methods section. You may consider the many possible resources to be reported as a table? Please also clarify what information will be used to derive your dataset and which is simply reported for referencing and to clarify the importance of your study. Also the PTFs are not clear why they are given extensively in the introduction. I suspect that this can become one subsection in the methods and again detail the ones you really used and give the other references as brief as necessary.

***We removed most of the dataset descriptions and web links from the introduction and rather focused on giving broader context and limitations of current fine resolution soil mapping.***

Methods:
As stated above, Figure 1 should come rather early. As you have done, the methods then follow the four steps. And as proposed earlier, I suggest to include the respective background information, references etc. in these subsections. So everything about the actual mapping, unit derivation, etc. is maybe 2.1, 2.2 is A etc.

All Figures should be included as a vector graphics if possible. Fig. 1 still holds v1.0 and other details which might be easy to clarify in a revision. You said USLE_k is omitted as intermediate parameter? I would also suggest to reduce the used symbols to a minimum and to give a legend. The 4 main boxes could simply get their respective headers. The dashed sub-boxes and speech bubbles might simply become the same shape as annotations? The resulting properties could be given as something like a bottom line?
***We simplified the figure and kept only the main workflow.***

In the methods, the subsections should really align with the workflow in Fig. 1 to avoid confusion. If necessary, details can be of cause extended in subsubsections. Make it easy for the readers to follow. Always trace what information is really used and what is further reference. Please try to give

it always the same pattern: What is needed, available sources, other examples, how did you derive the property and how will you evaluate the results. Please avoid jumps between sub-topics. I think it is all there but still a little hard to fit the puzzle. E.g. 2.2 is about texture. Texture can be encoded as fractions of sand, silt and clay (or finer textural classes) or as WRB name. The layers (2.2.3) are for me a different thing more related to the geometric questions of mapping.

2.3 starts off with reference to 2.4. Maybe this could be more content of the overall introduction when Fig. 1 is explained in detail? Then 2.3 is clear to derive topographic indices and you have room to also link to the method background why these indices are selected. Maybe it is not even necessary to use so many subsections?

***We removed the subsubsections and moved a lot of technical details to the supplements in order to keep the flow more straightforward for the reader.***

In 2.4 you again motivate the application of PTFs of 2.5 but the overall pathway could be given earlier to avoid confusion and to really concentrate on SOC, SOM and BD. Since you use a random forest, I would simply remain with RF as one way of machine learning. Avoiding to use the two terms as synonyms could give room for details and clarity. I think that it is important to also clarify how and when PTFs (which are often also ML e.g. Rosetta as ANN) are used. Are they really part of 2.4? 2.5 is very brief although I suspect it to be the very crucial step. Contrary to the former subsections which should be more concise, I think here could be a little more explanation e.g. why the AWC, FC, WP are important. Moreover, again some layers are defined. This time at fixed depths. I still struggle to understand the geometric idea of your database.

***We shortened the introduction and really try to focus only on PTFs and ML application used as PTF (such as in SoilGrids and EU-HydroSoilGrids) and only for our specific application domain.***

I think the final database including the respective names and encodings could be a further section after the methods. There can come all details about your naming conventions etc.

Results:
I understand the results to align best along the four subsections again. At the moment I really struggle to follow how the evaluation is done and if the maps are somehow reliable. Maybe you could use a few of the sampling areas from Fig. 2 plus a few other areas with information from other resources here? The maps of course look beautiful. But I have no idea if they achieve your goal for SWAT modelling.

***We reduced the number of figures and focused on less, but show the incredible detail as inset to demonstrate the high-resolution.***

Figure 9 is clearly result evaluation. But you refer to it only in the methods 2.2.4. I like the plot but cannot read the axes. I think this plot is too small. Maybe you can think of another form to give this reference? Moreover, I am a little puzzled about the continuous histograms in SoilGrids but the very few distinct bars in your database for texture. Also the other parameters do not really appear to resemble SoilGrids. I am not at all saying that SoilGrids is more correct. But you really have to discuss

why you come up with different values and why your data is trustworthy.

**We removed the pair-plot grids and focus on discussion and rather than 1:1 comparison as validation.**

Discussion:
Maybe you could move some of the open issues from the methods section to the discussion? I would expect a little more structure what is discussed here.

**We indeed removed parts from the introduction completely for focus and took a few sentences on soviet vs WRB/USDA particle size to discussion.**

Again, I am sorry to return the manuscript before sending it to reviewers. I simply fear that we might enter an internal loop of rather structural deficits which obscure to dive to the actual matter of your work. I hope you can easily follow my suggestions and that you find it helpful. In case of any questions, please contact me. We can even quickly arrange a video-call.

All the best.
Conrad

**On behalf of all authors,**

**Thank you very much for your guidance. We apologize for the hard-to-follow changes, as we have drastically restructured and re-written large sections of the manuscript, it is almost a new submission. We hope to have fulfilled your expectations and look forward to hearing back.**

**Alexander Kmoch**

[revised manuscript text omitted]

---

## Author Response (AR4)

**Referee report 1:**

GENERAL COMMENTS:

The authors restructured the manuscript and significantly rewritten several parts. This way several topics became easier to understand, most importantly the description on extracting soil texture and soil type, performance of bulk density estimation. Authors added Figure 1 to the manuscript which clearly summarize how the dataset was derived, this is very useful. The manuscript became more focused after authors decreased technical details and excluded those parameters which were not completely finalized, e.g.: soil fertility, organic horizon thickness. The reviewed manuscript is concise and easy to follow. There are some minor issues - listed under the specific comments - which could be considered/solved before publication.

Authors: We thank the referee for detailed comments. We address the specific comments point by point.

SPECIFIC COMMENTS:

P1 L13: something similar could be added: … 20 eco-hydrological variables on soil, topography and land use for Estonia …

added
P1 L15, P2 L30, Figure 1, P16 L13: … EU-SoilHydroGrids …

edited
P1 L23: … parameters include soil layering, soil texture (clay, silt and sand content), coarse fragments and rock content of the soil layers within the soil profiles.

rephrased
P1 L24-25: … predicted soil variables related to water and carbon cycle …

edited
P2 L4: .. where detailed soil survey data is available.

edited
P2 L21: … (approximately 1 km resolution) …

edited
P2 L23-24: … soil data, among others for sand, silt and clay content … organic carbon content …

edited
P2 L27: … measured soil profile data to train the model …

edited
P2 L31: … and pedotransfer functions (PTFS) trained on European samples.
The last sentence of the paragraph can be deleted.

Edited and deleted

P3 L3: … soil organic carbon content (SOC) …

Edited

P3 L4-5: monitor SOC changes … we didn't find the correct place (P3 vs P2? It seems there might have been a shift in the referred to page numbers by referee 1?

P3 L6-7: the sentence staring with "Very few" might not be true for the temperate region. Please rephrase the sentence.

edited

P3 L15: the paper of Abbaspour et al. (2019) (https://www.nature.com/articles/s41597-019-0282-4) could be mentioned as well.

added

P3 L22: .. at any given location in Estonia … ? or in the Baltic region?

Not clear where the comment can be pin-pointed.

P4 L1: … soil type, layering of the soil profile), texture …

edited

P4 L1-4: the sentences starting with "We derived" and "High resolution" seem to be repetition, might be deleted.

Not a repetition. We derived soil type, texture and layering without PTFs as a first step, then we use PTFs for SOC BD Ksat using derived sand silt clay etc.

P4 L4: … We employed … (without "thus")

edited

P4 L4-5: … additional soil variables …

edited

P4 L5: … filed survey soil data …

added

P4 L17-20: The sentence starting with "The subsequent sections" might not be that informative, the reader will see it, thus could be deleted.

We restructured based on ref2 comment

P5 Figure 1: in the block of LIDAR: … per soil unit (polygon)…

We don't think this would be correct.

P5 Figure 1: in the block of EU-SoilHydroGrids: … Saturated hydraulic conductivity …

edited

P5 Figure 1 caption: might be rephrased similarly: … , input datasets and derived eco-hydrological modelling parameters on soil, topography and land use.

edited

P5 L5: … based on observed …

*edited*
P5 L7: … intensive filed surveys …

*edited*
P5 L10-11: … with soil quality assessment (primarily fertility, rockiness, water regime, texture, erodibility).

*edited*
P5 L11: … field soil surveys were …

*edited*
P6 L3: the meaning of "textural status" is not clear, please rephrase it.

*rephrased*
P6 L14: … following attribute fields …

*We decided not to edit, as we believe this is more style choice by the reviewer.*
P6 L18: … fields of soil type …

*In special databases they are called attributes, not fields. We decide to keep the wording.*
P6 L23: Something similar could be added: Therefore an approach was developed to extract numerical soil information that is described under section 2.2.

*We are not sure where this is located, P6 doesn't have a L23.*
P9 L2-3: does it mean that uncertainty of organoleptic texture determination was checked based on laboratory data? If yes, please add how many samples was used for this analysis. If the 5% accuracy comes from another study, reference would be needed.

*This is referring to P5? We don't have a citation for that, and we don't have a specific number of samples. Some data is currently not accessible from the Soviet archives. However, we refer to the intense field surveys over several decades (cf. introduction and background on soil map sections), where several 1000 of samples were definitely analysed in laboratories.*
P9 L7: the following is a repetition, thus can be deleted: "and lack of silt data in the Soviet system"

*We disagree, the second mention of Soviet system is specific new fact.*
P9 L11: in Table 3 HUMUS is also included, please add its meaning and explanation on why it has been introduced under texture class.

*That is correct. We assume that this is possibly a data entry error wrongly entered into the texture attribute instead of the "humus" attribute that describes the organic litter layer, which usually is not accounted as soil layer. There are extremely few instances in fact, there are less than 90 occurrences (out of 845000).*
P10 L18-21: the sentence starting with "We compared" is not clear, please rephrase it.

*It is not clear where this is referring to: P7 and P10 have only a sentence that starts with "We compared"*
P10 L24: it is not clear how supplemental material can be accessed.

*The supplemental material is clear cited with a DOI which points to a Zenodo repository.*
P10 L27: … WRB reference soil group.

*The Estonian soil type classification is not identical to WRB. We explicitly distinguish carefully throughout the manuscript.*

P12 L2: … are drained per mapped unit … is it correct?

yes

P12 L3: … drainage regime …

edited

P12 L18-19: please consider the following modification: "We computed organic carbon content in % soil weight, and dry bulk density in Mg/m³ or g/cm³." Please clarify here for which soil layer these soil variables were computed: only for the topsoil (layer1) or for all the layers.

For all layers where the number of layers is defined based on the soil layering. We describe that throughout the manuscript.

P12 L22: It is not true that RF does not require preliminary hyperparameter tuning at all. Please rephrase the sentence. If the default values of the algorithm are used, than RF will not be sensible for parameter tuning. Please check it in the literature.

We agree, hyperparameter tuning as such has a measurable effect in RF. The wording preliminary tuning is misleading. Edited/rephrased

P12 L25: Was measured soil organic carbon content available for the topsoils or for the whole profiles? Was time series data available?

At different depths, but mostly top 30 cm, typically no time series

P14 L2: … we added two additional …

edited

P14 L3: … Saturated hydraulic conductivity (Ksat) is a quantitative …

edited

P14 L6-7: The sentence starting with "We develop two" is repetition can be deleted.

Edited/rephrased

P14 L11-15: the following belongs to results, please move these there: 1) the sentence starting with "Table 3 demonstrates" and 2) Table 3. In Table 3 please add the meaning of all abbreviations.

Moved.

P15 L9: … related to the reference groups of the World Reference Base and USDA texture descriptions.

rephrased

P17 Table 4: please add description of the following parameters: rock1-4, soc1-4, bd1-4, k1-4 awc1-4. For these only unit is given.

edited

P18 L2-3: the part of the sentence starting with "the soil experts" is not clear please rephrase it.

edited

P18 L20: please note that "only for forest soil samples" is in bold, please format it.

changed

P18 L20: please add what was compared to the observed value to compute RMSE and nMAD. Is that the extracted sand, silt and clay content that is characteristic for the texture classes?

We do not have enough sample data for fine earth fractions to make a generalising claim.

P18 L20-22: please move the sentence starting with "We calculated" under Materials and methods and add the equation of RMSE and nMAD there.

edited

P18 L23-28: please put these into a sentence and add number of samples used to compute RMSE and nMAD values.

Added, 84

P19 L3-5: The sentence starting with "For example" is not clear, please rephrase it.

rephrased

P19 L8: The sentence starting with "We also" is repetition, can be deleted.

edited

P19 L11: These might be the performance of the built RF model, is it correct? Please rephrase it in the manuscript.

We have difficulties in addressing this request, it is not clear to what it is aiming?

P19 L12-14: please put these into a sentence. Please add what "oob score" shows.

Oob score is a RF specific training metric, similar to "leave-one-out", but much faster. It relates identifying observations directly to trees in the forest. The higher the oob score the better the training score.

P19 Figure 4: please:

- add number of samples in the caption of the figure,

- add unit of x and y axis on the two plots and full labels.

Observations and predictions are simply a count/number. We updated the plots.

P19 caption of Figure 4: … predicted soil organic carbon content for the a) training …

adjusted

P20 L7-10: please put it into a sentence and add number of samples that was used to calculate RNSE and nMAD values.

P20 L14: … grasslands and forest based on RMSE… is it correct?

P20 Table 5: please add number of samples in a separate column. It might be enough informative to keep only the following: minimum, maximum, mean, median, std, nMAD and RMSE.

edited

P22 L8: … (clay, silt, sand content, …

edited

P22 L10: … saturated hydraulic conductivity …

edited

P22 L17: … holds the potential … is it correct?

correct

P22 L22-24: do you mean: to study temporal changes of soil and hydrological processes in Estonia? Please rephrase the sentence starting with "Furthermore,"

rephrased

rephrased

P22 L25: the beginning of the sentence starting with "In combination" is not clear, please rephrase it.

rephrased

P22 L30-31: What do you mean by "comparative validation"? It is not clear what kind of data was used from SoilGrids.

We used sand silt clay and rock content in technical supplement.

P22 L32-P23 L1: it is not mentioned in the present manuscript that raster datasets were compared to polygon-based data. Please clarify this sentence.

We moved it to the the technical supplement.

P22 L15: … Kõlli et al. (2009) … it could be considered that for peat special PTFs – derived specifically for peats – would be needed to compute their BD and soil hydraulic properties. Please describe it more why you had to assign sand, silt and clay content for the peat soil.

As we don't have any appropriate sampling data. The claim is for contiguous hydrological coverage. This needs to be verified through actual hydrological modelling. K_Sat values based on texture are corresponding with the literature for PEAT as described in the manuscript.

Under the website of EstSoil-EH (https://zenodo.org/record/3473290): EstSoil-EH_v1.0_attribute_fields.txt: please recheck the unit of the fields, e.g.: unit is not given in the case of attributes related to soil depth.

We have updated the deposit to https://zenodo.org/record/4068069

**Referee report 2:**

We thank the referee 2 for the valuable comments and tried to address the points raised in the manuscript comments here point by point.

P2 L23-25: edited, soil profiles replaced with specific examples

P3 L2: SOC inputs, deleted as of request from ref1

P3 L5, L16: edited

P3 L22-23: edited accordingly, moved higher and added 2.1 motivation

P4 L4: removed fully – just observed data

P5 L27: diameter

P6: fine earth fractions and coarse fragments

P6 L10: changed to: soil data coverage as little interrupted as possible

P7 L26: edited

P8 L3: represents (edited)

P8 LL4-5: deleted

P8 LL8-10: Paragraph/line feed to indicate a new thought (TRI description)

P8 LL11: considered and edited

P8 L21: edited

P9 LL7-8 edited

P9 L12: An alvar is a type of calcareous grassland, a distinct biological environment based on a limestone plain with thin or no soil and, as a result, sparse grassland vegetation. Description added.

P10 L8: edited

P11 L4: fine earth fractions is desired wording

P11 LL6-7: moved to results, added unit mm/h

P11 L11: edited, duplicate removed

P11 L13: edited

P13 L2: edited

P13 LL2-3: rephrased

P13 table edited

P14 L10: sentence deleted/rephrased.

P14 L16: edited

P14 L25: nMAD edited

P15 LL3-6: moved to discussion

P15 L14: Scikit-learn Python cited

P16 L13: added units

P16 L15: rephrased

P18 L3: We decided to not sub-divide the section, as the larger part of deals with the discussion of potential issues and specific characteristics for many of the variables, not only SOC and BD. In the main use case is outlined in the first two paragraphs and how the availability of this dataset now improves on the current situation.

P18 LL11-13: split and rephrased

P18 LL15-17: split and rephrased

[revised manuscript text omitted]

---

## Author Response (AR5)

**Editor comments:**

Thank you for the tremendous work you have invested. I think it was worth the effort. I am very happy to accept your contribution to our special issue.

I just spotted a couple of minor technical issues, I ask you to resolve before handing the MS to the copy editing and typesetting.

Authors: Thank you very much for the comments. We address the specific comments point by point.

SPECIFIC COMMENTS:

P3L20: RF is used for the first time here. I suggest to simply drop it. I still find the list of topics in the paper difficult to comprehend due to the shift in sentence structure. Moreover this paragraph sets off to give the structure of the paper but only drafts the next section 2. Pls. check.

Authors: We decided to remove the paragraph as to not confuse the reader. Alternatively spelling out that we derive the parameters in section 2, describe and evaluate the results in section 3 and discuss in section – we found to be superfluous.

P3L25: There appears to be a reference error of Latex.

Authors: fixed

P14L11: the last number is with comma as thousand separator. I do not know what is the house standard at copernicus but I think it is given in the author guidelines. Pls. check.

Authors: Actually, no explicit statement is to be found in the guidelines. Very large numbers are supposed to be written out like 1 Billion elements, but a 1043 was written as a plain number. We decided to stick with a space as thousand separator for consistency and removed the comma.

Fig 5: There are comma instead of dots as decimal separators...

Authors: fixed

P21L18: improvement > improved?
Authors: fixed

[revised manuscript text omitted]